# Multi-objective numerical optimization of 3D-printed polylactic acid bio-metamaterial based on topology, filling pattern, and infill density via fatigue lifetime and mass

**Ali Dadashi**, **Mohammad Azadi** *

Research Laboratory of Advanced Materials Behavior (AMB), Faculty of Mechanical Engineering, Semnan University, Semnan, Iran

* m_azadi@semnan.ac.ir

**Data Availability Statement:** All relevant data are within the paper and its Supporting Information files.

## Abstract

Infill parameters are significant with regard to the overall cost and saving material while printing a 3D model. When it comes to printing time, we can decrease the printing time by altering the infill, which also reduces the total process extent. Choosing the right filling parameters affects the strength of the printed model. In this research, the effect of filling density and infill pattern on the fatigue lifetime of cylindrical polylactic acid (PLA) samples was investigated with finite element modeling and analysis. This causes the lattice structure to be considered macro-scale porosity in the additive manufacturing process. Due to the need for multi-objective optimization of several functions at the same time and the inevitable sacrifice of other objectives, the decision was to obtain a set of compromise solutions according to the Pareto-optimal solution technique or the Pareto non-inferior solution approach. As a result, a horizontally printed rectangular pattern with 60% filling was preferred over the four patterns including honeycomb, triangular, regular octagon, and irregular octagon by considering the sum of mass changes and fatigue lifetime changes, and distance from the optimal point, which is the lightest structure with the maximum fatigue lifetime as an objective function with an emphasis on mass as an important parameter in designing scaffolds and biomedical structures. A new structure was also proposed by performing a structural optimization process using computer-aided design tools and also, computer-aided engineering software by Dassault systems. Finally, the selected samples were printed and their 3D printing quality was investigated using field emission scanning electron microscopy inspection.

## 1) Introduction

The functionality of a fabricated component relies on the combination of properties and geometry, achieved through a series of fabrication techniques [1]. Additive manufacturing (AM) processes offer advantages such as complex 3D geometries and flexible construction of layered structures [2]. Tailoring the build formulation allows for gradual changes in

**Funding:** The authors would like to acknowledge the financial support of the Iran Small Industries and Industrial Parks Organization (ISIPO) for this project under grant number of 23607.

**Competing interests:** The authors illustrate that they have no competing financial interests or personal relationships that could have appeared to influence this work.

composition or lattice structures [3, 4]. For instance, Kumar et al. [5, 6] compared multi-blended and hybrid blended PLA matrix in 3D-printed tensile specimens, finding superior mechanical and morphological properties in the multi-blended matrix, reflecting the growing interest in exploring the potential of additive manufacturing.

AM techniques enable the fabrication of metamaterials with unique properties, such as negative Poisson ratio and high strength [7–9]. These metamaterials exhibit extraordinary functions in acoustics, electromagnetics, and mechanics [10]. Bio-metamaterials, designed with geometrical structures at smaller scales, mimic the properties of living tissues and are applied in orthopedic biomaterials for bone tissue regeneration [11]. Although the feasibility and reliability of these processes are still recognized as a challenge [12]. Wu et al. [13] developed a novel mechanical metamaterial with lateral expansion, validated through simulations and experiments. Agrawal et al. [14] proposed a robust topology optimization method for negative Poisson ratio metamaterials, ensuring stable designs with low variation.

Researchers have utilized topology optimization (TO) and additive manufacturing (AM) to enhance the design and fabrication of structurally optimized cellular structures [15–16]. The integration of artificial intelligence (AI) and AM has introduced a new design principle, enabling efficient structure and component design through machine learning techniques [17–21]. Qian et al. [22] employed an artificial neural network-based inverse design method, demonstrating improved efficiency and reduced training data requirements for designing architected composite parts. Their approach showcased superior performance compared to other machine-learning approaches.

In order to further increase the mechanical properties of parts fabricated by AM, designing and optimizing approaches have been introduced and developed at various scales. Specifically, on the macroscale, structural optimizations including shape, size, and topology optimization techniques could be considered for guiding the component design [23, 24]. Between these methods, polymeric composites can achieve their maximum potential by AM-driven TO [25]. Manufacturing constraints such as minimum size and connectivity limit the optimization according to the manufacturing ability [26]. The anisotropic material property, besides manufacturing constraints, is another parameter that should be applied during the process of designing and optimizing [27]. A transversely isotropic material model and solid anisotropic material with penalization coupled with an AM-driven topology optimization technique were proposed by Li et al. [28, 29]. Due to the layered fabrication process, they achieved mechanical improvement in stiffness and strength because of taking advantage of anisotropy. Wu et al. [30] presented a novel method for generating simultaneously optimized solid shell and porous infill in the context of maximum stiffness topology optimization. A material interpolation model, upon which the compliance is minimized, unified the resulting intermediate density distributions while all the fibers follow the direction of the principal stress.

3D printing process parameters have a significant impact on the mechanical properties of printed parts, including printing temperature, speed, filling density, pattern, layer thickness, material properties, surface roughness, and layer thickness [31–33]. Researchers have conducted several studies to further clarify these effects [34, 35]. Manufacturing 3D-printed products requires a clear understanding of the mechanical characteristics of 3D-printed components [36]. Sood et al. [37] found that increasing the layer thickness in ABS 3D-printed components improved impact and bending properties, but had an inverse relationship with tensile performance. Fernandez Vicente et al. [38] examined the influence of different infill patterns on the tensile performance of ABS 3D-printed parts. They found that ABS components with line and honeycomb infill patterns exhibited higher tensile strength compared to those with a rectilinear infill pattern. Baptista and Guedes [39] investigated the impact of infill density and patterns on the static and fatigue characteristics of scaffolds. Lower infill density

resulted in higher yield stress and showed no significant fatigue damage after 3600 load cycles. Moreover, the influence of infill line distance on the compressive elastic behavior of 3D printed circular samples was investigated by Pepelnjak et al. [40]. They observed a significant difference in the first cycle response and an alteration of stiffness due to elastic and plastic deformation under cyclic.

Fountas et al. [41] examined the performance of different swarm-based evolutionary algorithms in optimizing single and multi-objective problems related to additive manufacturing, specifically fused deposition modeling (FDM). The results showed that algorithms do not perform equally when applied to different optimization problems, and the quality of Pareto non-dominated solutions is an important indicator for multi-objective problems. Yodo and Dey [42] introduced multi-objective optimization methods based on evolutionary algorithms to optimize FDM process parameters, leading to Pareto-optimal solutions for decision-makers.

So far, the literature has been reviewed and by summarizing the presented materials, it can be said that researchers have focused on the following topics:

- The extraordinary properties of meta-materials [8, 9, 13]

- Providing optimization algorithms to improve mechanical behavior [17–23]

- Designing optimized structures in different dimensions [26–30]

- Applying additive manufacturing conditions in the optimization process [31–39, 43–46]

- Introducing Pareto-optimal solutions for decision-makers [41, 42]

The article discusses the multi-objective numerical optimization of 3D-printed polylactic acid (PLA) bio-metamaterial. The optimization takes into account the factors of topology, filling pattern, and infill density in order to improve the fatigue lifetime and mass of the material. Therefore, the novelties of the study are as follows,

- The use of multi-objective optimization to find the optimum value of topology, filling pattern, and infill density for 3D-printed polylactic acid bio-metamaterial.

- The consideration of both fatigue lifetime and mass in the optimization process.

- The use of field emission scanning electron microscopy (FESEM) images to investigate the dimensional accuracy of 3D-printed components.

## 2) Material and methodology

### 2–1) Material properties

Manufacturing parameters have a significant impact on material response, particularly for polymers [47, 48]. For example, it was reported that the Poisson ratio is influenced by infill density [49]. However, in this study, the material properties of filament were used and the investigation of other parameters was not included. Density, elastic module, yield strength, ultimate tensile strength, and Poisson ratio were obtained from technical datasheets provided by the material manufacturer. Based on Table 1, these properties are typically reported by the manufacturer based on standard testing methods [47]. Though, endurance limit, which refers to the maximum stress amplitude that a material can withstand for a given number of cycles without failure, cannot be obtained from technical datasheets and requires experimental testing. The endurance limit used in this study was obtained experimentally from the reference [48]. It should be noted that YouSu brand polylactic acid (PLA) filament, was used in component fabrication for 3D printing. The properties of 3D printed parts can differ from those of

**Table 1. Mechanical properties of 3D-printed PLA [47, 48].**

| Characteristics | Unit | Amount |
|---|---|---|
| Density | g/cm$^3$ | 1.252 |
| elastic modulus | MPa | 1280 |
| Yield strength | MPa | 70 |
| Ultimate tensile strength | MPa | 73 |
| Poisson ratio | - | 0.36 |
| Endurance limit | MPa | 7.3 |

the raw filament material. For example, while the density of PLA filament was typically reported as 1.24 g/cm$^3$ in manufacturer datasheets based on the ASTM D792 standard test method, 3D printed parts can have a different density due to their meta-structure. Additionally, the tensile module and yield strength of PLA filament are reported as 3.5 GPa and 60 MPa, respectively, based on ASTM D1238 standard test method. However, 3D printed parts may exhibit different values for these properties due to the complex interplay of factors such as infill percentage, layer height, printing speed, and other printing parameters. The endurance limit for 95% probability of survival and determined in terms of maximum stress and extrapolated at 2 million cycles to failure cycles, is considered equal to 0.1 final stress [48].

The compressive and tensile behaviors of PLA are usually considered to be the same, but some studies have focused specifically on the compressive behavior of PLA. These studies aim to better understand how PLA responds to compressive forces and how it can be used in applications where compression is a dominant mode of loading, such as in bone tissue engineering and orthopedics [50, 51].

## 2–2) Numerical simulation

Structural optimization is an iterative technique that could help the refinement of designs. Then, a result of the proper-designed structural optimization could be a lightweight part, which has durable properties. An integrated computer-aided design (CAD) and computer-aided engineering (CAE) workflow by Dassault systems was proposed (Fig 1).

The use of additive manufacturing enables the fabrication of different patterns with varying mechanical properties. The filling parameters have a direct effect on the response of the meta-materials to external loadings. For instance, a recent study [52] on metal functionally graded gyroids employed three design methods (thickness graded, size graded, and uniform) to provide a map of the mechanical properties of stainless-steel scaffolds. In the proposed study, we selected five infill patterns, including honeycomb, rectangular, triangular, irregular octagon [52], and regular octagon [52] to investigate their effect on fatigue properties and mass, and the results suggest that further investigations on other patterns may be necessary, to compare the fatigue behavior. The infill patterns used in this study are shown in Fig 2.

Based on the flow chart, a geometric model of the structure should be initially prepared in CAD software, including SolidWorks or Catia. Considering Fig 3, filling densities of 40%, 60%, 80%, and 100% were also designed in the section of a cylinder with an outer diameter of 9 mm and a height of 78 mm, by SolidWorks software. To achieve desired filling densities these geometries were arranged next to each other at different distances. In addition, two solid layers are considered as a perimeter with a width of 0.4 mm for each layer [53].

A finite element model is fabricated in SIMULIA- ABAQUS for stress analysis besides the topology optimization. To calculate bending stress according to the rotating-bending fatigue testing method [54], the samples were subjected to a 10 N load at the free end of a cantilever

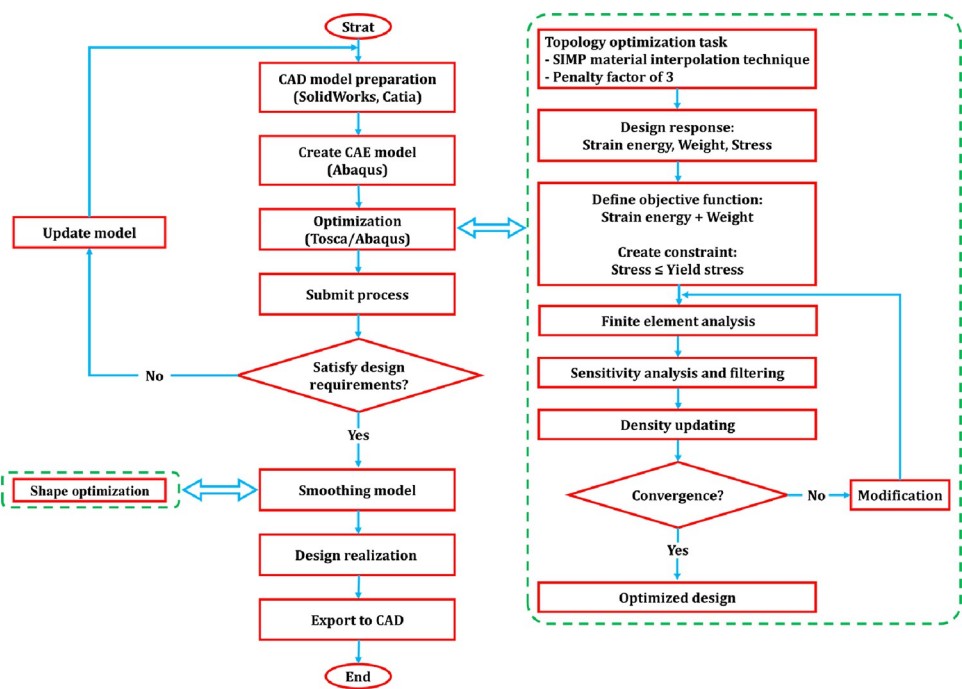

**Fig 1. The workflow of topological designing using Dassault systems software.**

which is clamped at the other end; therefore, the displacements of the supported end are constrained in three directions (Fig 4(A)). In the rotating-bending test, a common method used to evaluate the high cycle fatigue behavior of materials, the specimen is subjected to cyclic loading

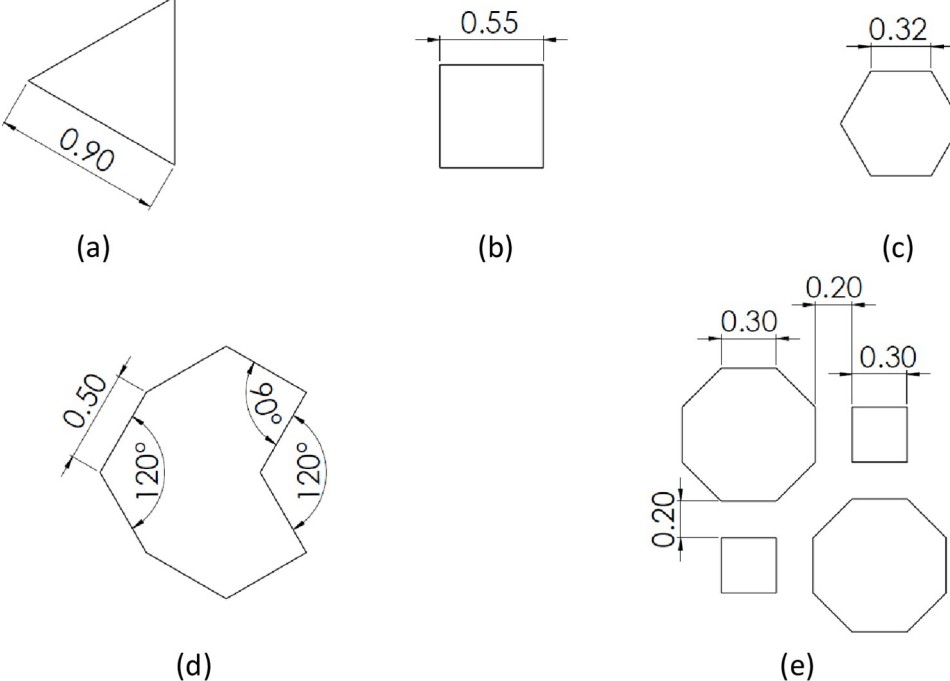

**Fig 2.** Different filling patterns: (a) triangular, (b) rectangular, (c) honeycomb, (d) irregular octagon, and (e) regular octagon.

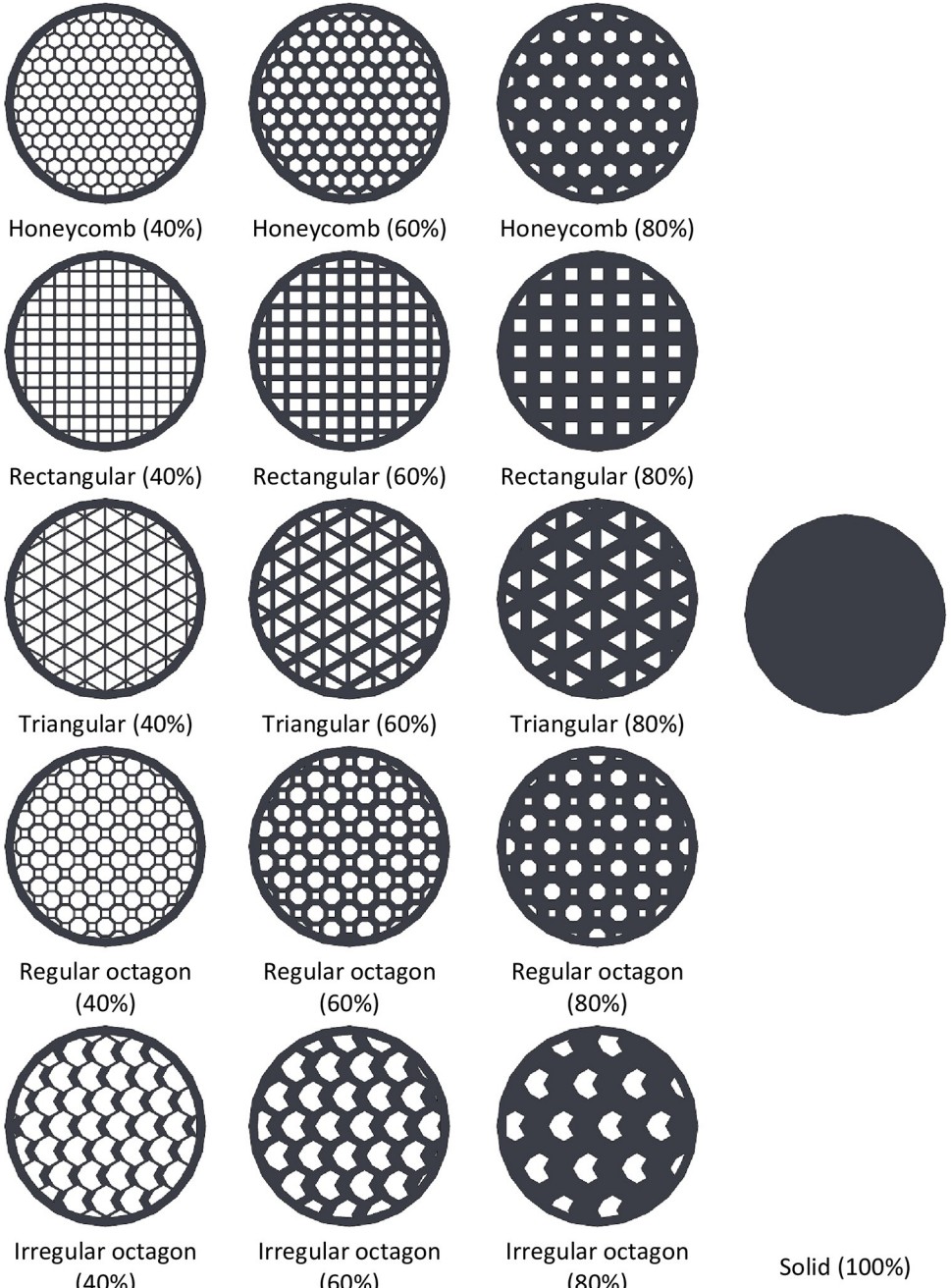

**Fig 3. The cross-section of metamaterial structures with varying infill densities.**

through rotational motion. During the rotating-bending test, the applied stress levels are typically kept below the yield strength of the material. This ensures that the deformation experienced by the specimen remains within the elastic range ($<70$ MPa [47]), where the material can fully recover its original shape after each cycle. The focus is on assessing the material performance to withstand cyclic loading without undergoing plastic deformation or failure. In this regard, static, general simulations were performed in Abaqus/Standard. Regarding the mesh size, 8-node linear hexahedral solid elements with reduced integration (C3D8R) are

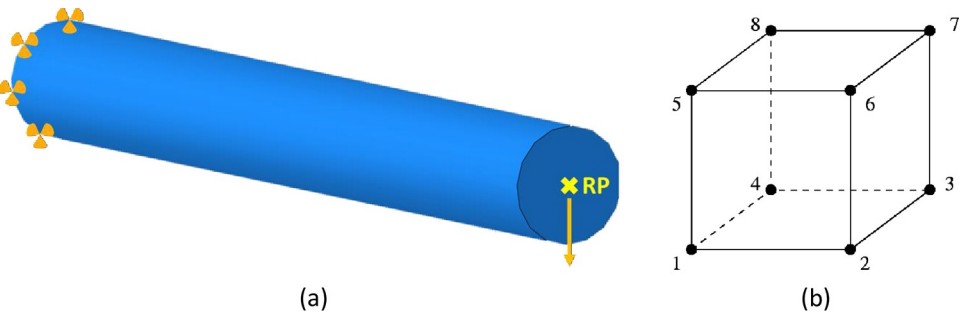

**Fig 4.** (a) The illustration applied load plus boundary conditions and (b) the C3D8R element.

considered with the global size of 0.7 mm to provide optimal accuracy and decrease CPU time and storage requirements (Fig 4)B) ([55].

The study aimed to analyze the impact of infill patterns and densities on printed part performance, without modeling of the layered structure or the print process. Therefore, the nozzle diameter was not directly considered in the numerical analysis. However, certain areas in the model had thicknesses below 0.7 mm, which required special consideration to prevent the element distortion. Therefore, two control parameters were utilized, including minimum size control which set to 0.1 of the global size and a curvature control parameter with a maximum deviation factor of 0.1. These adjustments ensured that elements in thin regions maintained high-quality meshing while adhering to the desired accuracy levels. The chosen control parameters were recommended by the ABAQUS software algorithm. Accordingly, Table 2 presents the minimum element length and the number of elements for various infill patterns and filling densities. Mesh sensitivity analysis was also conducted to assess the impact of varying element sizes on the performance of models. It was observed that the von-Mises stress became stable once the element size reached 0.7. As the element size reduced, the stress values showed fluctuations and variations until they reached the critical value of 0.7. Beyond this point, a further reduction in element size did not significantly affect the stress levels, indicating that the mesh

**Table 2. The number of elements in different models.**

| Fill pattern | Infill (%) | Minimum size (mm) | Number of elements |
|---|---|---|---|
| Honeycomb | 40 | 0.079 | 163947 |
|  | 60 | 0.139 | 93129 |
|  | 80 | 0.136 | 124653 |
| Regular Octagon | 40 | 0.065 | 118690 |
|  | 60 | 0.104 | 13961397 |
|  | 80 | 0.162 | 131262 |
| Irregular Octagon | 40 | 0.116 | 101232 |
|  | 60 | 0.180 | 93936 |
|  | 80 | 0.315 | 26418 |
| Rectangular | 40 | 0.090 | 102897 |
|  | 60 | 0.196 | 51837 |
|  | 80 | 0.400 | 22644 |
| Triangular | 40 | 0.033 | 149850 |
|  | 60 | 0.116 | 73704 |
|  | 80 | 0.109 | 90021 |
| Solid | 100 | 0.576 | 17538 |

had reached a point of convergence. This finding suggested that an element size of 0.7 can provide a balance between the accuracy and computational efficiency for the specific analysis of the model.

## 2–3) Topology and shape optimization

Topology optimization is a proper method for designing that provides engineers with a powerful code in order to evaluate the solution space and enhance the designing reliability for any structures, in macro- and micro-scales. A wide range of various techniques exists such as the solid isotropic material with penalization (SIMP) or the rational approximation of material properties (RAMP). Then, the tool carries out several iterations to find a consecutive answer by redesigning and simulating the structure. As the RAMP interpolation scheme is suitable for dynamic problems, the SIMP algorithm was used in this study.

For this topology optimization, the superior technique is the SIMP approach, which estimates an optimal distribution of the material under boundary conditions, loadings, manufacturing constraints, and any other requirements for the performance and properties.

The main technique in topology optimization is to discretize the solution domain into a grid of finite elements, which is entitled isotropic solid microstructures. Then, two events occur either the element is materially filled or the element becomes a void without a material (material-removing). This issue could be written as follows in mathematics by the density distribution of the material ($\rho$), which is a discrete binary value [55],

- $\rho_{(e)} = 1$ for material requiring,

- $\rho_{(e)} = 0$ for material removing.

Then, since the material density is a continuous value; therefore, the elastic module varies continuously in all elements, as the following equation [55],

$$E(\rho_e) = \rho_e^p E_0 \tag{1}$$

In this power law for each element (e), $\rho_e$ is the relative density of the material, $E_0$ is the elastic module, and finally, $p$ is a factor for the penalty which shows the contribution of elements with intermediate values of the density to the total stiffness. Based on the literature the numerical experiments indicate that a penalty factor value of $p = 3$ is suitable in the SIMP method for the topology optimization [55, 56].

As known, when the elastic module reduces, the element stiffness decreases. Based on the SIMP approach, the stiffness could be calculated as follows [55],

$$K_{SIMP}(\rho) = \sum_{e=1}^{N} [\rho_{min} + (1 - \rho_{min})\rho_e^p] K_e \tag{2}$$

In this formulation, $\rho_{min}$ is the minimum relative density and $K_e$ is the matrix for the element stiffness, where $N$ is the element number. As a note, through the optimization iteration, a sensitivity analysis is done in order to find the influence of the density variation on the objective of the problem.

In the present study, the solid sample model was also used to find a topologically optimized structure. Then, the objectivities and the constraints were defined for the optimization in the topological design technique by simulation-based SIMULIA-Tosca, with a sensitivity-based optimizer to find an optimized trend for designing.

The optimization objective was to minimize the structure mass (the mass summation for all elements, $\Sigma W_e$) and also the strain energy (the compliance of a structure which is defined as

the summation of the strain energy for all elements, $\Sigma u^t K u$. In this case, $u$ is the displacement and $K$ is the matrix to the global stiffness). This job was done under the stress constraint, which must be less or equal to 70 MPa which is the yield stress of polylactic acid (PLA). It should be noted that because of reduced relative density during the topology optimization, interpolation is required to calculate the stresses of elements, so the scaled element centroidal von-Mises stress of the design area is used in the optimization process [55],

$$Stress = \max \left| \frac{(\sigma_{vMises})^2}{(f(\rho_i)\sigma_y)^2} . \sigma_y \right| \qquad (3)$$

where, $\sigma_{vMises}$ is the element centroidal von-Mises stress, and $\rho_i$ demonstrates a parameter for interpolating the stress in the element. This issue leads to a reduction in the current relative density due to the topology optimization. Moreover, the reference stress, $\sigma_y$ is given as follow [55],

$$\sigma_y = \min \left\{ \alpha_1 \max \left| \frac{(\sigma_{vMises})^2}{(f(\rho_i)\sigma_{ref_1})^2} . \sigma_{ref_1} \right| + \alpha_2 \max \left| \frac{(\sigma_{vMises})^2}{(f(\rho_i)\sigma_{ref_2})^2} . \sigma_{ref_2} \right| \right\} \qquad (4)$$

$\alpha_1$ and $\alpha_2$ are the weighting factors and $\sigma_{ref}$ is the reference stress calculated during the initial optimization iteration [55].

As an important note, a mesh smoothing process was used by an additional shape optimization. This job was to have better quality for elements and also to minimize the value of von-Mises stress, as a minor modification. In other words, shape optimization was utilized in order to minimize the stress concentration in the finite element model, as a stress homogenization process [55].

Failing to properly realize the optimized design may result in a product that does not perform as intended or that is not feasible to manufacture, and therefore, it is crucial to carefully consider a design realization step in the topology optimization process to achieve a feasible component design concerning the manufacturing constraints. Therefore, for further post-processing development, some modifications were performed manually inspired by the geometry resulting from shape optimization, used to redesign a new part.

## 2–4) Fatigue performance

To evaluate the fatigue lifetime, Eq (5) was used [57]:

$$\sigma_a = \sigma_f' (2N_f)^b \qquad (5)$$

in which $b$ is the fatigue strength exponent, and $\sigma_f'$ is the fatigue strength coefficient. Besides, $\sigma_a$ is the stress amplitude which is equal to the maximum stress obtained from numerical simulation. Now the fatigue lifetime, $N_f$, can be calculated.

These parameters are material constants and differ notably when the printing conditions change. Nevertheless, $b$ and $\sigma_f'$ are provided in Table 3 for two print directions based on the previous studies by the authors [58]. In this table, fatigue tests were conducted at each stress

Table 3. The parameters of the material including fatigue properties [58].

| Material / Print direction | | All data | | | Mean value | | |
|---|---|---|---|---|---|---|---|
| | | $b$ | $\sigma_f'$ | $R^2$ | $b$ | $\sigma_f'$ | $R^2$ |
| PLA | Horizontal | -0.2850 | 321.27 | 0.9680 | -0.2880 | 334.29 | 0.9805 |
| | Vertical | -0.2880 | 180.25 | 0.9573 | -0.3010 | 207.90 | 0.9993 |

level three times. Then, an S-N diagram was provided that fitted all the data. Moreover, the average of the results at each stress level was calculated and then again, another S-N diagram was provided. Although these material properties in Table 3 are not exactly as same as the material behavior of the fabricated samples in this work; they could be an initial prediction for the lifetime estimation. Besides, in this work, the infill density and the filling pattern were considered to study. These parameters could have effects on the stress value. This influence was also investigated by microscopic images, as to check the 3D printing quality.

## 2–5) Two-objective optimization

A quantitative technique for solving two-objective optimization was used to find the optimized value of filling parameters. Therefore, using these optimization objectives leads to having a minimal mass and a maximal fatigue lifetime. In this case, three sets of objective functions were considered. The first one uses the mass ($M$) and fatigue lifetime ($N_f$). The mass can be calculated with the production of material density and the volume of each specimen by fatigue lifetime is also obtained from Eq 5. Mass reduction ratio and fatigue lifetime reduction ratio compared to the solid sample according to Table 4 are the second set of objective functions. The third set includes the distance from the optimal point (maximum mass loss and least lifetime reduction), $D$, and the sum of mass and fatigue lifetime, $S$.

In Table 4, $M$ is mass and $N_f$ is the fatigue lifetime. $\Delta N_f\%$ and $\Delta M\%$ are changed in fatigue lifetime and mass compared to the solid sample, respectively. $N_{f(solid)}$ and $M_{solid}$ are the fatigue lifetime and mass of the solid sample, $S$ is the sum of mass and fatigue lifetime changes, and $D$ is the distance from the optimal point. Due to the decrease in mass compared to the solid sample, the mass change is considered negative.

Considering different objectives in the Pareto front diagram, the performance-price ratio was used as an optimization reference. These objectives are similar to the fatigue lifetime and the mass (Set 3 in Table 4). Higher fatigue lifetime means better performance and lower mass means lower cost.

Accordingly, all stages of this analysis could be summarized based, as follows [59],

**Step 1:** Sorting the solutions in the diagram of the Pareto front by the objective ($f_1$) in ascending behavior [59].

**Step 2:** Considering the average line slopes for the average variability, based on objective functions ($f_1$ and $f_2$). The formulations for these slopes ($k_1^{(m)}$ and $k_2^{(m)}$) are as follows [59],

$$k_1^{(m)} = \frac{1}{2}\left(\frac{f_2^{(m)} - f_2^{(m-1)}}{f_1^{(m)} - f_1^{(m-1)}} + \frac{f_2^{(m+1)} - f_2^{(m)}}{f_1^{(m+1)} - f_1^{(m)}}\right), m = 2, 3, \ldots, M-1 \tag{6}$$

$$k_1^{(m)} = \frac{1}{2}\left(\frac{f_1^{(m)} - f_1^{(m-1)}}{f_2^{(m)} - f_2^{(m-1)}} + \frac{f_1^{(m+1)} - f_1^{(m)}}{f_2^{(m+1)} - f_2^{(m)}}\right), m = 2, 3, \ldots, M-1 \tag{7}$$

**Table 4. The combination of objective functions.**

| Set | $f_1$ | $f_2$ |
|---|---|---|
| Set 1 | max: $N_f = 0.5 \times \left(\frac{\sigma_a}{\sigma_f}\right)^{\frac{1}{b}}$ | min: $M = \rho V$ |
| Set 2 | min: $\Delta N_f\% = \frac{N_{f(solid)} - N_f}{N_{f(solid)}} \times 100\%$ | min: $\Delta M\% = -\frac{M_{solid} - M}{M_{solid}} \times 100\%$ |
| Set 3 | min: $S = \Delta N_f\% + \Delta M\%$ | min: $D = \sqrt{\left(\Delta N_f\%\right)^2 + \left(\Delta M\% + 100\%\right)^2}$ |

Let $M$ denote the number of Pareto non-inferior solutions, and let $x^m$ (m = 1,2,...,M) denote the set of these solutions. In the Pareto front, the slope at endpoints (m = 1 and M) are $k_1^{(1)}, k_2^{(2)}, k_1^{(M)}$, and $k_2^{(M)}$, which could be proposed as follows [59],

$$k_1^{(1)} = \frac{f_2^{(2)} - f_2^{(1)}}{f_1^{(2)} - f_1^{(1)}} \tag{8}$$

$$k_2^{(1)} = \frac{f_1^{(2)} - f_1^{(1)}}{f_2^{(2)} - f_2^{(1)}} \tag{9}$$

$$k_1^{(M)} = \frac{f_2^{(M)} - f_2^{(M-1)}}{f_1^{(M)} - f_1^{(M-1)}} \tag{10}$$

$$k_2^{(M)} = \frac{f_1^{(M)} - f_1^{(M-1)}}{f_2^{(M)} - f_2^{(M-1)}} \tag{11}$$

**Step 3:** Performing the sensitivity analysis between two average variabilities in a specific non-inferior Pareto solution. For this purpose, the sensitivity ratio is defined as $\delta_1^{(m)}$, which is proposed by a ratio of $k_1^{(m)}$ to $f_1$. Then, $\delta_2^{(m)}$ could be also considered as the ratio of $k_2^{(m)}$ to $f_2$. These formulations could be found as follows [59],

$$\delta_1^{(m)} = \frac{k_1^{(m)}}{f_1^{(m)}}, m = 1, 2, \ldots, M, f_1^{(m)} \neq 0 \tag{12}$$

$$\delta_2^{(m)} = \frac{k_2^{(m)}}{f_2^{(m)}}, m = 1, 2, \ldots, M, f_2^{(m)} \neq 0 \tag{13}$$

**Step 4:** Nondimensionalizing the sensitivity ratio for a better comparison. Supposing $\varepsilon_1^m$ and $\varepsilon_2^m$ are the results of $\delta_1^m$ and $\delta_2^m$, respectively, the non-dimensionalization leads to [59],

$$\varepsilon_1^{(m)} = \frac{\delta_1^{(m)}}{\sum_{i=1}^M \delta_1^{(i)}}, m = 1, 2, \ldots, M \tag{14}$$

$$\varepsilon_2^{(m)} = \frac{\delta_2^{(m)}}{\sum_{i=1}^M \delta_2^{(i)}}, m = 1, 2, \ldots, M \tag{15}$$

**Step 5:** Considering two values for the solution ($x^{(i)}$ and $x^{(j)}$) in the Pareto front diagram, if $\varepsilon_l^{(i)} > \varepsilon_l^{(j)} (l = 1, 2)$; then, $x^{(i)}$ dominates $x^{(j)}$ and $x^{(i)} > x^{(j)}$; else, $x^{(i)}$ becomes the non-dominant solution or non-inferior solution in the sensitivity analysis [59].

**Step 6:** Considering a bias degree for various objectives between the value of 0 and 1. Then, For $M^*$ as the solution number in the domain of $X^*$, $w_1^m$ and $w_2^m$ are defined as [59],

$$w_1^m = \frac{\varepsilon_1^{(m)}}{\varepsilon_1^{(m)} + \varepsilon_2^{(m)}}, m \in M^* \tag{16}$$

$$w_2^m = \frac{\varepsilon_2^{(m)}}{\varepsilon_1^{(m)} + \varepsilon_2^{(m)}}, m \in M^* \tag{17}$$

where,

$$w_1^m + w_2^m = 1, m \in M^* \tag{18}$$

For objective functions of $f_1$ and $f_2$, $(w_1)_{max}$ and $(w_2)_{max}$ have a strong value of the bias degree [59].

**Step 7:** Using Formulation (19), $\Delta\varepsilon^{(m)}$ (m = 1, 2, . . ., M) could be considered as [59],

$$\Delta\varepsilon^{(m)} = |\varepsilon_1^{(m)} - \varepsilon_2^{(m)}|, m = 1, 2, \ldots, M \tag{19}$$

**Step 8:** Lowering $\Delta\varepsilon^{(m)}$ to obtain $(\Delta\varepsilon)_{min}$, which is the best unbiased solution. Notably, in this condition, the values of $f_1$ and $f_2$ are acceptable [59].

$$(\Delta\varepsilon)_{min} = \min\{\Delta\varepsilon^{(1)}, \Delta\varepsilon^{(2)}, \ldots, \Delta\varepsilon^{(M)}\} \tag{20}$$

**Step 9:** End.

## 2–6) 3D printing

Printing some regions of a part before starting the main study is a method of optimally manufacturing specimens, thereby reducing the financial and time loss caused by the trial-and-error stage. The selected patterns, including honeycomb, rectangular, triangular, regular octagon, and irregular octagon with 60% infill were printed along with rectangular patterns with 40% and 80% of filling densities were printed as shown in Fig 5. The green lines in Fig 5

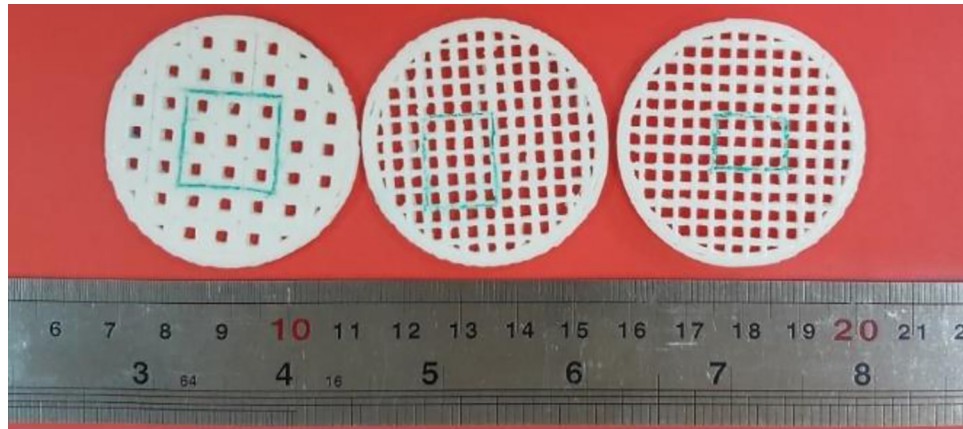

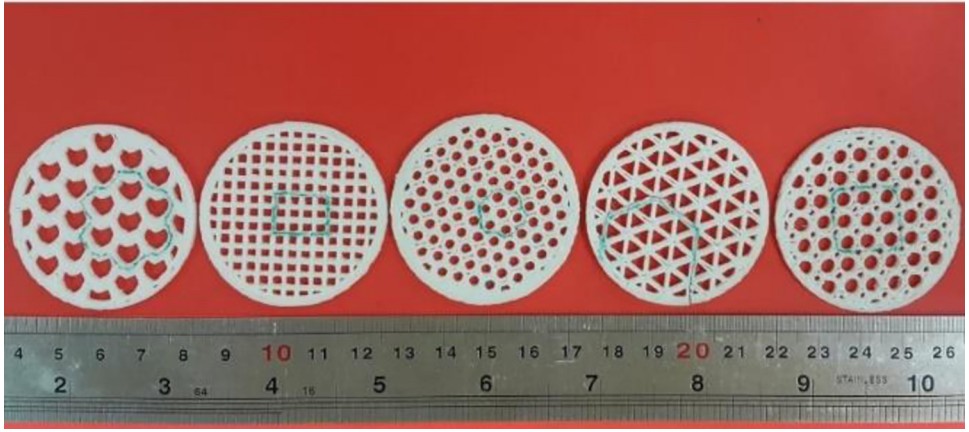

**Fig 5. The geometry of the 3D-printed specimens with different filling patterns and infill densities.**

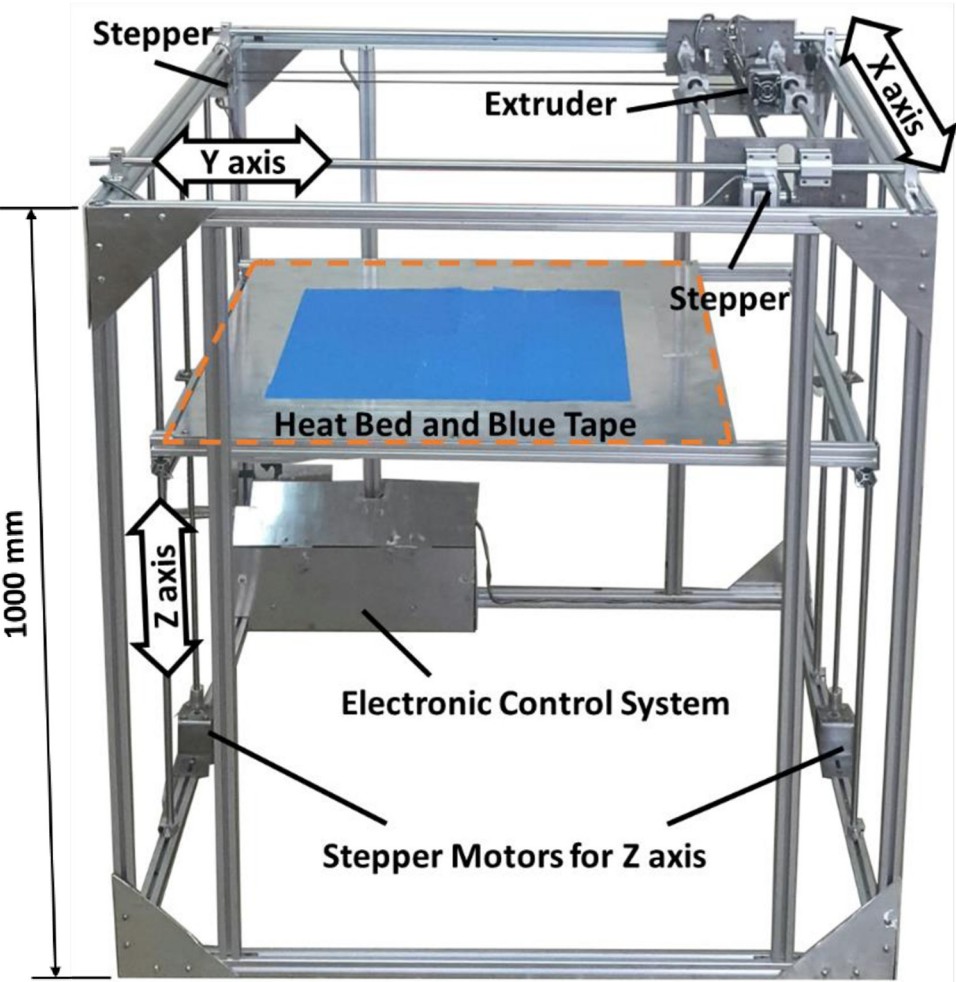

**Fig 6. The FDM 3D printer used for additive manufacturing of the specimens.**

represent the desired part of the print samples that was cut and evaluated through inspection in the FESEM.

In order to provide the specimens, an FDM 3D printer was developed in the research laboratory of advanced materials behavior (AMB), as shown in Fig 6. Table 5 shows the conditions under which all of the samples were printed. To ensure optimal results and utilize the capabilities of an industrial-duty 3D-printer machine, the samples were printed at a scale of 5 times larger than the original CAD models. One other reason is due to the machine limitations. This approach allowed to produce lattice structures with circular sections and a diameter of 45 millimeters, which would have been challenging to achieve with smaller scales. While this approach allowed for the successful printing of the specimens, it is important to investigate the potential effects of scaling on the accuracy of the geometrical features and the resulting mechanical properties in future investigations. Previous studies demonstrated clear size effects on the mechanical response of additively manufactured lattice structures [60]. 3D printing was done with white-colored PLA filaments, which diameter was 1.75 mm, by the YouSu Company. The samples were printed with 100% infill and no top and bottom solid layers printed, allowing for clear visualization of the print path.

**Table 5. The parameters of 3D printing [58, 61, 62].**

| Process variable | Operating condition |
|---|---|
| Nozzle diameter (mm) | 0.40 |
| Filament diameter (mm) | 1.75 |
| Layer height (mm) | 0.20 |
| Perimeter | 2 |
| Top and bottom solid layers | 0 |
| Print speed (mm/s) | 10 |
| Travel speed (mm/s) | 30 |
| Nozzle temperature (°C) | 210 (First layer) 180 (Entire layers) |
| Bed temperature (°C) | 25 (Room temperature) |

### 2–7) Inspections

A close examination of the samples was performed after printing. For this purpose, a conductive golden layer was applied on the samples by gold sputtering and inspected using a TESCAN (MIRA 3 LMU). Imaging was performed with magnifications of different magnitudes, including 15X, 22X, 27X, 30X, 80X, and 500X. The length of the vertices and width of lines were measured in the microscopic images to understand the accuracy of prints.

## 3) Results and discussion

### 3–1) Stress analysis

Fill pattern and filling density are critical conditions for the fabrication of AM components. To investigate the influence of these parameters on the performance of printed parts, specimens with different filling patterns and densities were modeled in ABAQUS software, as shown in Fig 4. Due to the great dependence of the stress analysis results on boundary conditions and loadings, especially the maximum stress, the constraints must be carefully set. Based on the results from stress analysis from FEM simulations, the maximum von-Mises stress occurs in the clamped region. As expected, there is a tensile state of axial stress at the top of the beam and a compressive axial stress state at the bottom. Thin walls between the patterns cause stress concentration, so the lower filling density cause higher stress. But the filling pattern affects differently in each filling percentage. As shown in Fig 7(A), honeycomb samples had higher von-Mises stress. However, irregular octagon, rectangular and regular octagon had the lowest stresses.

The comparison of weight reduction in this study is conducted by considering the calculated weight values, taking into account the theoretical density of the material and the measured volumes of the samples in the CAD model. By utilizing these parameters, the weight of the objects can be estimated. Based on the results shown in Fig 7(B), the mass of the honeycomb and rectangular lattice structures was nearly identical, with the rectangular pattern being slightly heavier. In addition, the triangular pattern exhibited less mass compared to the other three patterns. The irregular octagon, regular octagon, and rectangular patterns were the heaviest ones. However, it is clear that by decreasing filling density, specimens get lighter, and the triangular pattern has the lowest mass in all cases.

The stress distribution contours obtained from finite element simulation are summarized in Figs 8 and 9. As shown in stress distribution contours, the bending load generates compressive stress on the bottom side of the clamped end and tensile stress on its top side. However, in fatigue testing, the load is cyclic and changes the loading direction periodically, which leads to

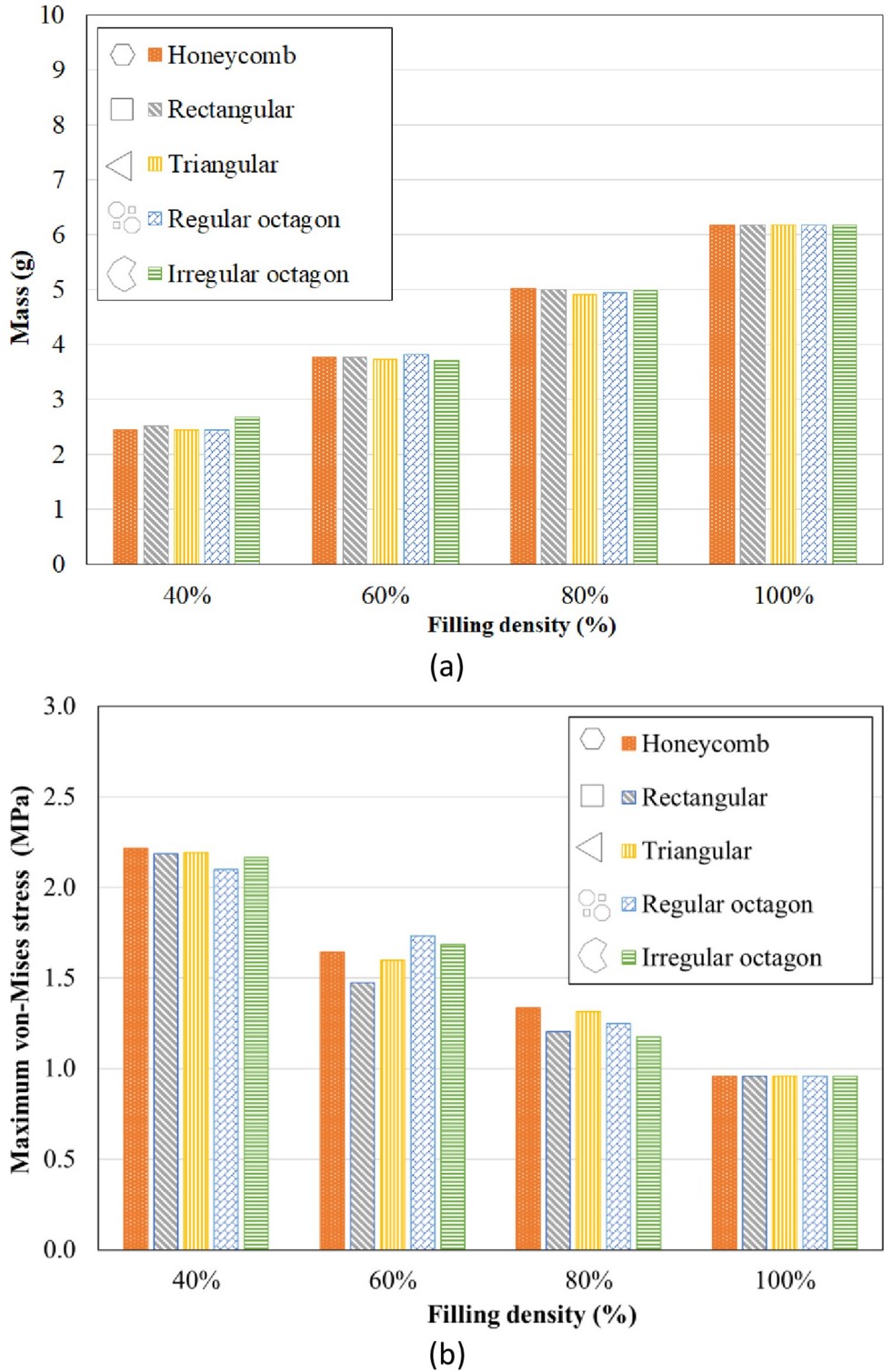

**Fig 7.** The influence of filling parameters on (a) the mass of structures and (b) the maximum von-Mises stress.

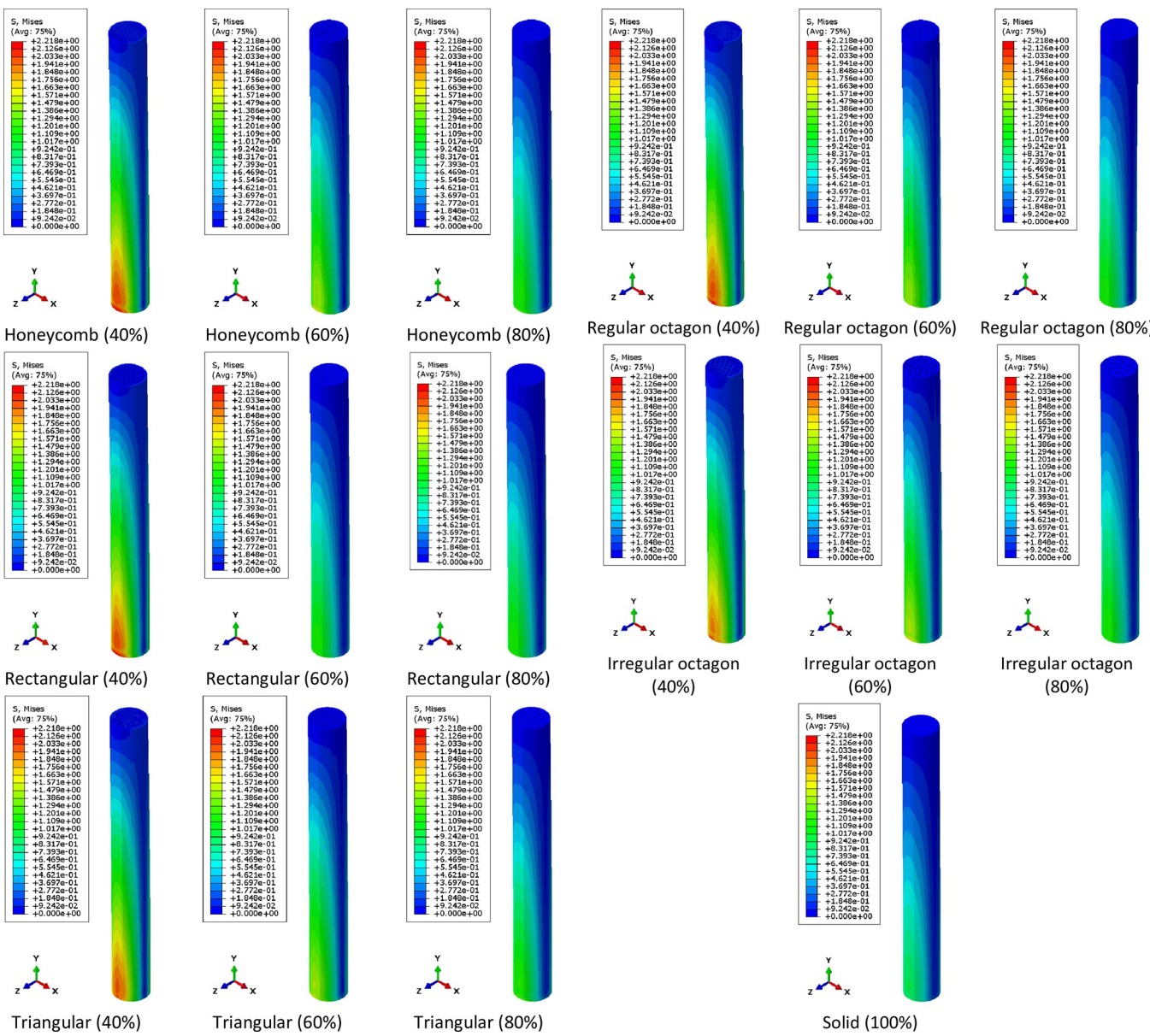

**Fig 8. The von-Mises stress distribution contours of metamaterial structures with varying infill densities.**

alternating regions of compressive and tensile stresses. Therefore, the specimens undergo both compressive and tensile stresses during fatigue testing, which can affect their durability and mechanical performance. Although the stress distribution in PLA specimens under cyclic loading cannot be measured experimentally, the fatigue behavior of the material is still worth investigating in future researches. This information can be compared to numerical results.

## 3–2) Two-objective optimization

To find the optimized filling parameters, a multi-objective optimization method proposed by Wang et al. [59], was used. First, based on the objective function proposed in Table 4, the distribution of the Pareto non-inferior solution set was obtained for each set. Using Formulations

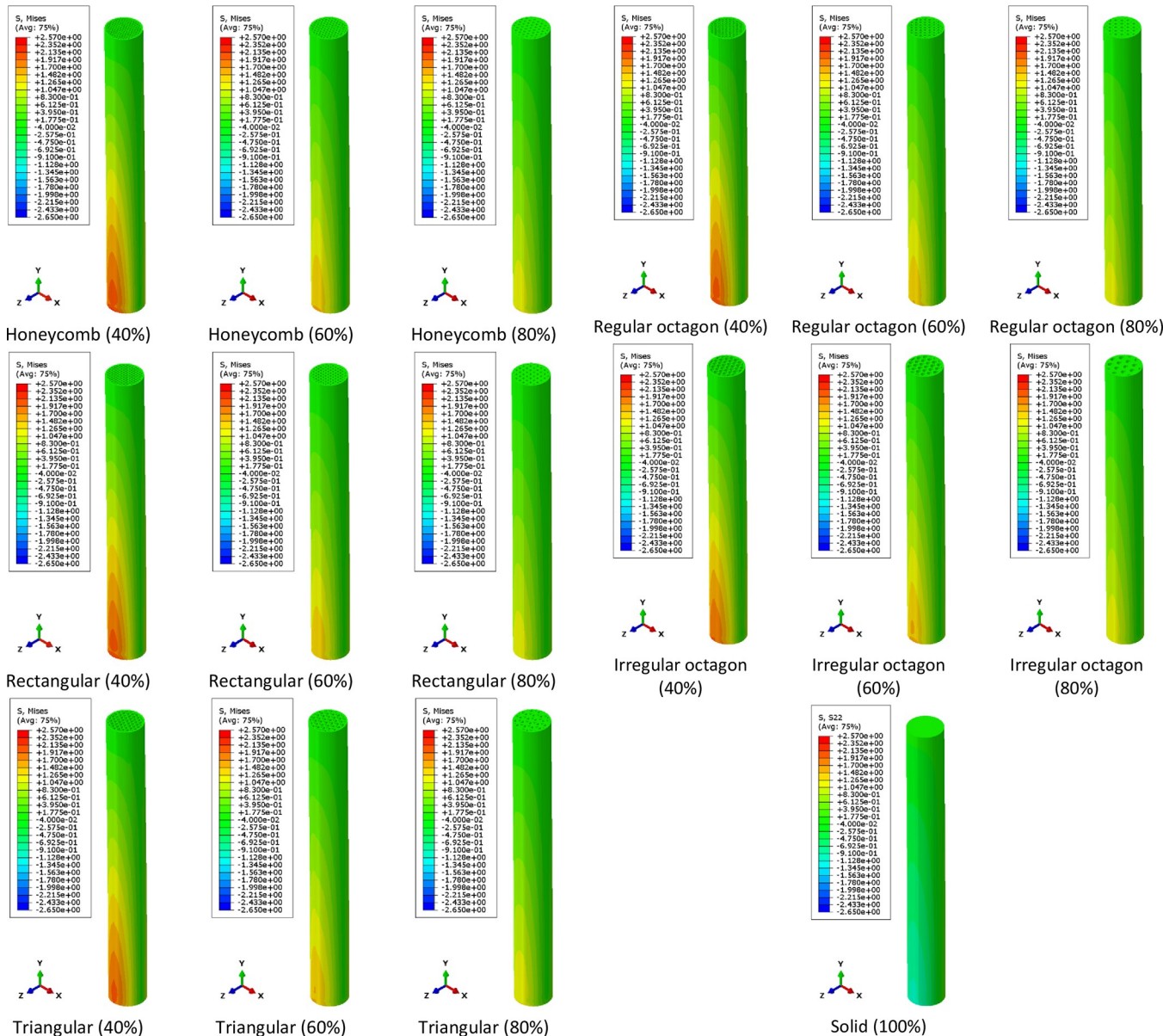

**Fig 9. The normal stress distribution contours in Y direction (σyy) contours of metamaterial structures with varying infill densities.**

(6)–(13), the sensitivity analysis on the non-inferior solutions of $f_1$ and $f_2$ was calculated. Then, based on Formulations (14) and (15), the sensitivity ratio of $f_1$ and $f_2$ was normalized. Formulations (16) and (17) were considered to find $w_1$ (for $f_1$) and $w_2$ (for $f_2$). Based on the dominance relationship of the bias degree of the 15 Pareto non-inferior solutions from each set of objective functions, eight solutions have been ruled out. The remaining solutions form the subset of non-inferior solutions based on the bias degree. Using Formulation (19), the values of Δε are obtained for each set of objective functions and presented in Tables 6 and 7 for horizontal and vertical print directions, respectively. The index of Δε in these tables indicates the corresponding objective function number. Therefore, $(\Delta\varepsilon)_1$, $(\Delta\varepsilon)_2$, and $(\Delta\varepsilon)_3$ are related to the first, second, and third set of objective functions. The minimum value of this parameter, $(\Delta\varepsilon)_{min}$, can be the superior unbiased solution, which could be acceptable for $f_1$ and $f_2$.

**Table 6. $\Delta\varepsilon$ corresponding to Pareto non-inferior solutions for vertical print direction.**

| No. | Filling density | Fill pattern | $(\Delta\varepsilon)_1$ | $(\Delta\varepsilon)_2$ | $(\Delta\varepsilon)_3$ |
|---|---|---|---|---|---|
| 1 | 40 | Honeycomb | 0.1095 | 0.0902 | 0.4380 |
| 2 | 40 | Rectangular | 0.3254 | 0.2972 | 0.4344 |
| 3 | 40 | Triangular | 0.3342 | 0.3029 | 0.4474 |
| 4 | 40 | Irregular octagon | 0.1166 | 0.1183 | 0.4177 |
| 5 | 40 | Regular octagon | 0.1085 | 0.0956 | 0.4645 |
| 6 | 60 | Honeycomb | 0.5889 | 0.1199 | 0.4561 |
| 7 | 60 | Rectangular | 0.6461 | 0.1046 | 2.3375 |
| 8 | 60 | Triangular | 0.4181 | 0.0665 | 2.3249 |
| 9 | 60 | Irregular octagon | 0.3591 | 0.0696 | 0.4842 |
| 10 | 60 | Regular octagon | 0.0293 | 0.0322 | 0.3795 |
| 11 | 80 | Honeycomb | 1.4354 | 0.6114 | 0.3521 |
| 12 | 80 | Rectangular | 1.3944 | 0.6027 | 0.0628 |
| 13 | 80 | Triangular | 0.0903 | 0.0394 | 0.0073 |
| 14 | 80 | Irregular octagon | 0.0735 | 0.0295 | 0.3310 |
| 15 | 80 | Regular octagon | 0.2226 | 0.0927 | 0.3872 |

In the case of vertical print direction, the values of $\Delta\varepsilon$ are listed in Table 6. The solutions of 5[th], 6[th], 7[th], 8[th], 9[th], 10[th], 13[th], and 15[th] from Set 1, 5[th], 6[th], 7[th], 8[th], 9[th], 10[th], 13[th], and 14[th] from Set 2 and 4[th], 5[th], 6[th], 9[th], 10[th], 11[th], 14[th], and 15[th] from Set 3 are dominated and therefore, they should be ruled out. Hence, the solution of 14[th], 11[th], and 13[th] are good results for vertical print direction and acceptable for both objectives in the first, second, and third set of objective functions, respectively. As it is obvious, by selecting different objective functions, the results will be different.

On the other hand, for longitudinal print direction, according to the data listed in Table 7, the 1[st], 5[th], 6[th], 7[th], 8[th], 9[th], 10[th], and 13[th] solutions from Set 1 are dominated. The 1[st], 6[th], 7[th], 8[th], 9[th], 10[th], 13[th], and 14[th] solutions should be ruled out from Set 2 based on the dominance relationship. The 1[st], 2[nd], 5[th], 7[th], 8[th], 11[th], and 15[th] solutions are also the remaining solutions from Set 3. Therefore, the 14[th], 15[th], and 2[nd] solutions meet Eq 20 and are determined to be

**Table 7. $\Delta\varepsilon$ corresponding to Pareto non-inferior solutions for horizontal print direction.**

| No. | Filling density | Fill pattern | $(\Delta\varepsilon)_1$ | $(\Delta\varepsilon)_2$ | $(\Delta\varepsilon)_3$ |
|---|---|---|---|---|---|
| 1 | 40 | Honeycomb | 0.1092 | 0.0893 | 0.0907 |
| 2 | 40 | Rectangular | 0.3264 | 0.2948 | 0.0461 |
| 3 | 40 | Triangular | 0.3353 | 0.3000 | 0.0281 |
| 4 | 40 | Irregular octagon | 0.1152 | 0.1178 | 0.0469 |
| 5 | 40 | Regular octagon | 0.1066 | 0.0940 | 0.0668 |
| 6 | 60 | Honeycomb | 0.5302 | 0.1176 | 0.0129 |
| 7 | 60 | Rectangular | 0.5815 | 0.1004 | 0.1055 |
| 8 | 60 | Triangular | 0.3729 | 0.0633 | 0.0934 |
| 9 | 60 | Irregular octagon | 0.3198 | 0.0670 | 0.0116 |
| 10 | 60 | Regular octagon | 0.0265 | 0.0328 | 0.0980 |
| 11 | 80 | Honeycomb | 1.3343 | 0.6003 | 0.0484 |
| 12 | 80 | Rectangular | 1.2958 | 0.5945 | 0.0632 |
| 13 | 80 | Triangular | 0.0839 | 0.0400 | 0.0390 |
| 14 | 80 | Irregular octagon | 0.0691 | 0.0289 | 0.0497 |
| 15 | 80 | Regular octagon | 0.2083 | 0.0932 | 0.0756 |

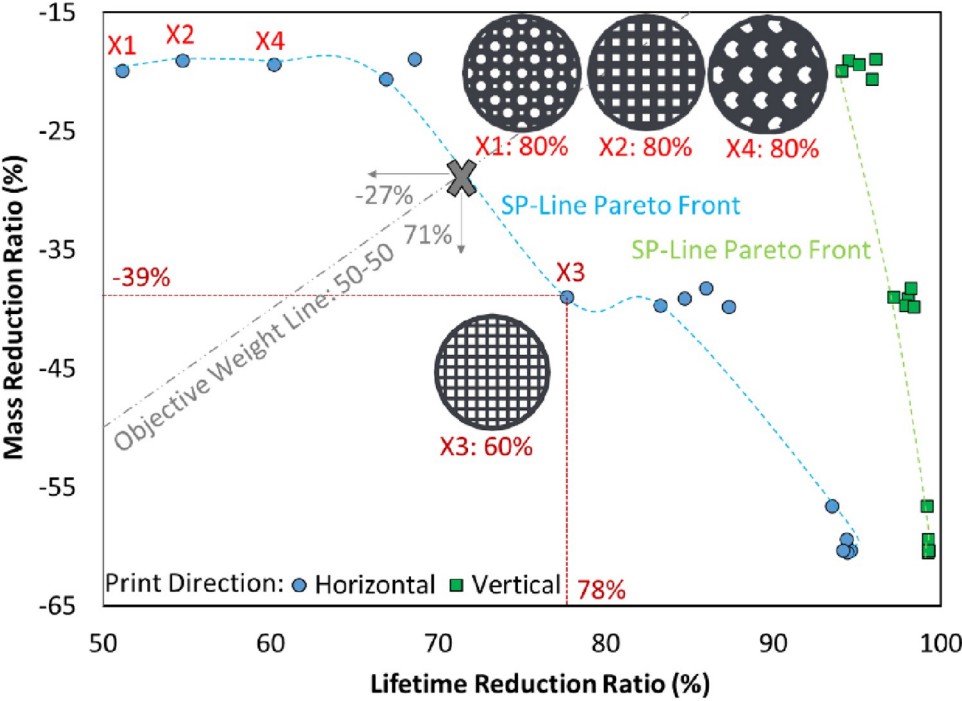

**Fig 10. The Pareto front for filling parameters.**

the unbiased and good solutions in the case of horizontal print direction for Sets 1, 2, and 3, respectively. Notably, the quantized values of the bias degree for objectives are also calculated.

The decision can be balanced and selected more efficiently according to the quantitative indicator. For example, if objective $f_1$ is preferred in the third set of objectives, the 7th non-inferior solution should be selected as the bias degree of the solution, which reached 0.8112 for $f_1$ and it is only 0.1888 for $f_2$. Moreover, if the objective of $f_2$ is preferred, the 1st non-inferior solution is selected as the bias degree of the solution reaches 0.7742 for $f_2$ and it is only 0.2258 for $f_1$.

In Fig 10, the Pareto front indicates that there was not much change in the fatigue lifetime of vertical printed specimens (green points) even with a severe reduction in mass. On the other hand, the horizontal orientation (blue points) resulted in layer lines being placed horizontally in the direction of the bending force, making it the optimal orientation for mechanical strength. Despite the need for support during printing, this orientation provides the best mechanical strength. This result is also obtained in the literature [58, 63], so horizontal print direction is selected in this study. It should be mentioned that horizontal printing improves mechanical strength due to aligned loading and better interlayer bonding, while vertical printing, with perpendicular load and layer-by-layer nature, may lead to reduced strength. This is while, both print directions maintain consistent structure from bottom to top layer [58].

In Fig 10, the point at which the 50–50 line crosses the Pareto front indicates a state of a 50% mass reduction and a 50% increase in the lifetime compared to the solid sample. According to the importance of mass in biomechanical applications, the rectangular pattern with 60% filling can be a good choice among the two samples that are closest to the point where the 50–50 line meets the Pareto front. Considering the interaction of objective functions, in the rectangular pattern with 60% of the filling density, the fatigue life has decreased by only 22%, despite the 39% reduction in mass. In addition, this sample has the shortest distance to the optimal point in the results of two-objective optimization by selecting the third set of objective

functions (in Table 4) with an emphasis on the second objective. Based on this, fatigue samples should be printed with a 60% rectangular pattern.

The obtained results in the literature have also illustrated that the infill pattern through 3D printing influenced the mechanical responses of AM structures. For example, the triangular infill pattern with 40% infill density in PLA achieved a maximum peak force in low-velocity impact testing compared to the quarter cubic, grid, and tri-hexagon patterns [64]. The optimal filling pattern for the nylon matrix with a reinforcement under the uniaxial tension-tension fatigue test was triangular, with 20% of the filling percentage [65]. Moreover, the triangular shape had a superior tensile strength in the composites, reinforced with continuous carbon fibers and chopped carbon fibers (Onyx) [66]. The rectilinear pattern presented the best mechanical performance, having a high elastic module, tensile strength, and impact resistance, compared to the grid, honeycomb, and triangle [67]. The specimens manufactured through a filling pattern of honeycomb with 75% of the filling density could be preferred to the rectilinear samples to increase the fatigue lifetime of PLA-wood composite structures [53].

Vigneshwaran and Venkateshwaran [68] found that changing the filling percentage (in naturally available wood flour reinforced with 30–90% of biodegradable PLA) multiplied the tensile performance and also the stiffness by factors of 2.0 and 1.5, respectively for wood-PLA biocomposites. According to their results, the printing pattern, whether hexagonal, rectangular, or triangular did not find to affect the final characteristics of bio-composites. The rectangular-filled nylon structure illustrated the maximum value for tensile properties compared to triangular- and hexagonal-filled components. On the other hand, the triangular-filled part had the minimum value of tensile properties, while the hexagonal-filled sample demonstrated a lower value for tensile performance than the rectangular-filled part [69]. The highest value of strength and lowest value of deformation are related to the irregular octagon structure and under 10 N compression load compared to the regular octagon under 10 N compression load [52]. As a result, depending on the material and the conditions of use, the performance of filling parameters is different and should be considered in the design domain.

## 3–3) Numerical results and optimization

The topology optimization procedure was done in ABAQUS with the SIMP technique. Fig 11 shows the resulting plot of this process. According to this figure, a mass of 1.1767 g and a total strain energy of 0.056 J were achieved after 50 cycles. As a result of the optimization efforts, the desired optimized geometry was obtained. A threshold of the ISO (MAT_PROP_NORMA-LIZED in the legend of Fig 12) value is generally used for regulating, which the finite elements are displayed. This value is utilized to obtain the positions on the finite element edges, where the new nodes are made and must be between zero and unity. Larger values lead to a model with a smaller volume. The determined design is depicted in Fig 12. Notably, it should emphasize that the proposed geometry only demonstrates the elements with a density higher than 20% to maintain the continuity of components.

The optimized geometry in Fig 12 was reached after 50 design cycles. An animation is also included in S1 Fig that shows the topology optimization process developments during each design cycle, providing a visual representation of the design process. Such visual support presents a better understanding of the design evolution and the optimization steps taken to arrive at the final design. The optimized geometry has a unique characteristic in its design—a quadruple thread-like pattern surrounded by a shell. This pattern is evident throughout the structure and adds a distinct visual aspect to the overall design. The quadruple thread-like pattern with its shell provides an optimized structure with improved mechanical performance by reducing mass and enhancing strength in specific locations. The thread-like pattern serves as a

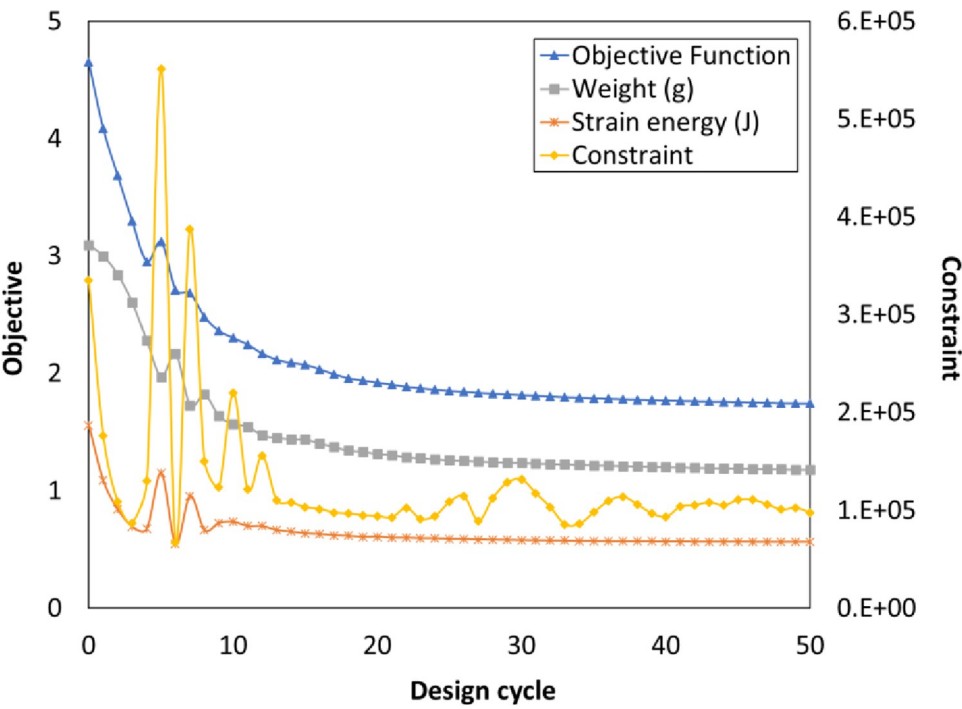

**Fig 11. The variations of design responses through design cycles.**

unique solution in topology optimization and can be customized to meet specific design requirements. Accordingly, the design realization step was performed for further considerations of manufacturability. This could be a crucial step in topology optimization since it is the stage where the optimized design is translated into a manufacturable and practical product. In this step, the abstract and optimized design obtained from the topology optimization process is transformed into a tangible, functional product that can be manufactured. The design realization step involves several important tasks, including geometry clean-up, and preparation of the final design for manufacturing. In this step, the design must be refined to ensure that it meets all design requirements and constraints and can be manufactured using the desired materials and processes. The realized design, which is presented in Fig 13, is believed to be conceptually feasible due to rotational symmetry. This design approach has potential applications in various industries, such as biomedical to produce stents, where reducing mass and increasing strength are critical considerations. The printed specimen in Fig 14 exhibits the desired geometry and structural features as intended in this design. This physical realization served as a tangible representation of the proposed model and validated the effectiveness of the present design process. It provided concrete evidence that the optimized design can be successfully translated into a physical object through 3D printing.

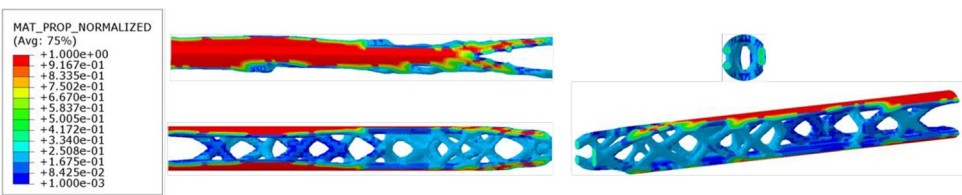

**Fig 12. The extracted geometry from topology optimization.**

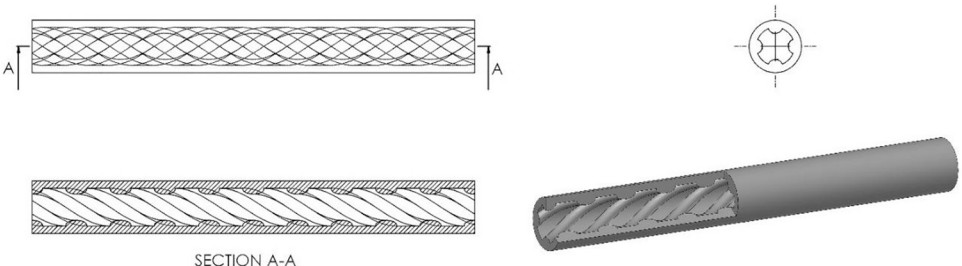

**Fig 13. The geometry of the designed sample based on the result of topology optimization.**

Figs 15 and 16 present the results of the static analysis of the optimized components. Fig 15 presents three geometries after (a) topology optimization, (b) shape optimization, and (c) design realization steps. The figure provides a visual depiction of how the optimization process affected the stress distribution of the component and how the optimized design was implemented in the final product. The maximum von-Mises stress was 97.08 MPa, which was observed after the topology optimization and located at the point shown in Fig 15(A). The results represent an output of a complex geometry from the optimization process. However, experimental investigations are necessary to confirm the reliability and accuracy of the optimization procedure. As it can be seen, the obtained geometry proposed several sharp edges and also some irregular surfaces. This issue is related to mesh refinement under boundary conditions.

Thus, a post process should be done based on smoothing and repairing the previous operations on such a sharp surface, by a shape optimization in order to minimize the von-Mises stress. As a result, maximum stress homogenized and decreased to 44.30 MPa after 15 design cycles (Fig 15(B)). The geometry after the design realization step is the final product of the optimization process. This geometry represents the optimized component that has been

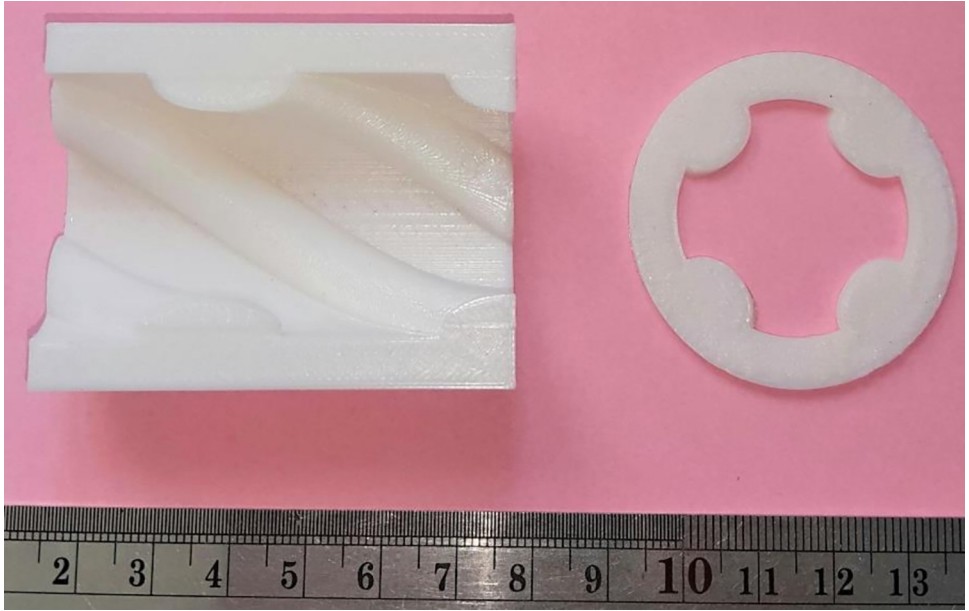

**Fig 14. The designed sample printed based on the result of topology optimization.**

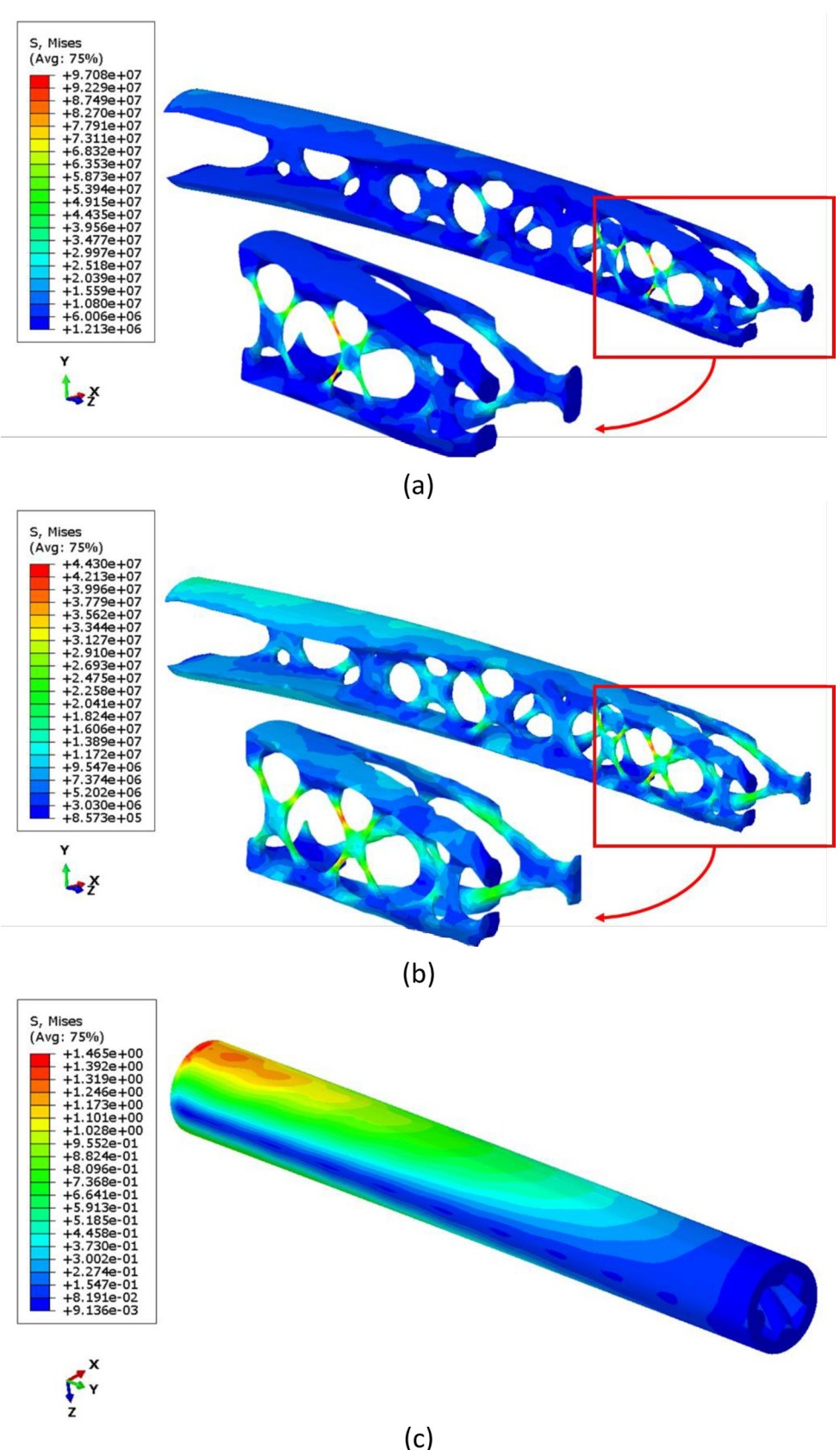

(a)

(b)

(c)

**Fig 15.** The von-Mises Stress distribution in the optimized component after (a) topology optimization and (b) shape optimization (c) design realization.

manufactured using 3D printing technology. The optimized component was designed based on the results of the multi-objective optimization, which took into consideration factors fatigue lifetime and mass. Despite the increase in mass compared to the topology optimized sample, the maximum von-Mises stress greatly reduced, based on Fig 15(C). With a mass of 3.56 g, this sample is lighter than the samples with 60% of filling density. However, due to less stress, it tolerates more cycles. The design realization step is the final step in the optimization process and the optimized component is now ready to be used in applications that require its unique properties. Fig 16 displays the stress distribution along the component axis in optimized geometries. The contour lines in the figure accurately represent the stress distribution, including stress concentration regions, mirroring the pattern observed in the von-Mises stress. Additionally, the figure distinctly reveals areas with compressive stresses.

Table 8 compares the results obtained from the various optimization processes. This table provides a quantitative evaluation of the impact of the optimization process on the mass of components, maximum von-Mises stress and fatigue lifetime based on the printing direction. The comparison allows the assess the effectiveness of the optimization process and draw conclusions about the optimal values. This information can be useful for engineers and designers in future 3D printing projects.

### 3–4) Morphological characteristics

The desired part of the print samples, which are marked with green lines in Fig 5, has been cut. The printed samples were cut with a fretsaw along marked lines for the analysis, ensuring precise removal without affecting surrounding areas. Then, the cut samples evaluated by inspection in the FESEM, and then examined for dimensional and geometrical accuracies. In order to assure adequate electrical conductivity for FESEM observation, samples were coated with a golden layer. The imaging process was performed with different magnifications, including 15X, 22X, 27X, 30X, 80X, and 500X. These magnifications were employed to capture various levels of details, allowing for a more comprehensive evaluation of the studied samples. The printed samples had acceptable overall compliance of the shape with designed geometries, according to low magnification images (Fig 17). However, it is also obvious that the regularity of the rectangular pattern was the best, while the regular octagon pattern was the most irregular. Easy to be noticed by the eye, the line spacing is significant in the honeycomb pattern. In addition, with higher fill percentages, the spacing between lines decreases.

As illustrated in Fig 18, voids could be observed at the turning points of the extruder. According to previous studies, this void enhances by increasing the temperature of the extrusion. An incorrect control of the temperature through filament depositing can affect the adhesion between 3D-printed layers and also the filament sides along the filling pattern through the raster (Fig 19)A() [70]. In other words, the air gap between the rasters enhances by increasing the deposition temperature. Notably, the raster width has a higher value at lower temperatures and therefore, it effectively reduces by enhancing the deposition temperature [71].

As observed in Fig 19(B), some small spherical-shaped defects may be generated during the fabrication of the filament, which can negatively impact the mechanical properties. According to the literature [72], the moisture significantly affected the filament behavior. Additionally, the extrusion feed rate plays a critical role in determining the quality of printed parts, as shown in Fig 19(C). If the extrusion feed rate is not balanced with the extrusion temperature and printing speed, inconsistent fusion can occur, resulting in defects. Another parameter that can

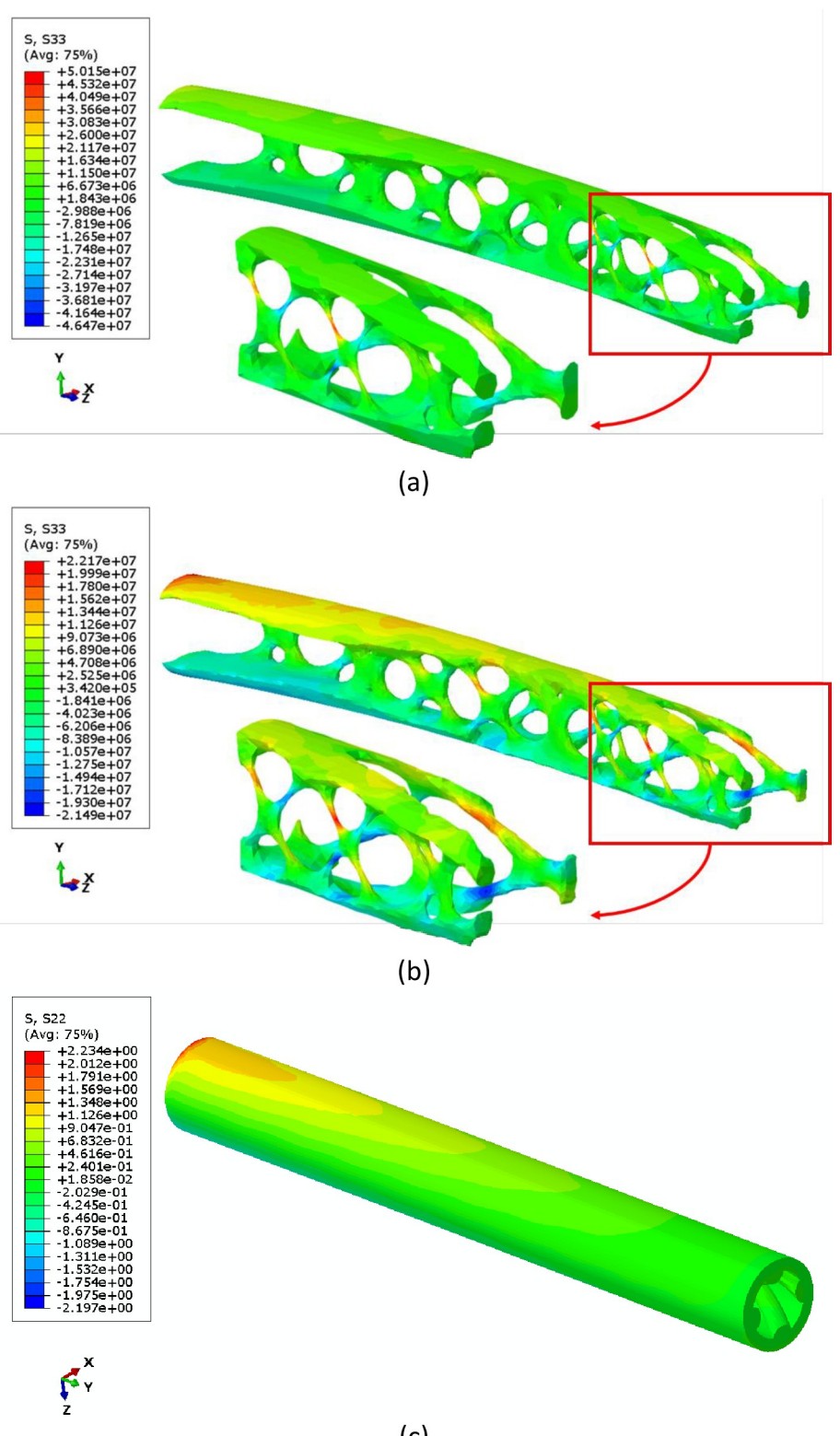

**Fig 16.** The stress distribution in the direction of cylinder axis in the optimized component after (a) topology optimization and (b) shape optimization (c) design realization.

**Table 8. The comparison of the results through the optimization processes.**

| Optimization process | Mass (gr) | Maximum von-Mises stress (MPa) | Fatigue lifetime (cycle) | |
|---|---|---|---|---|
| | | | Horizontal | Vertical |
| Topology | 1.18 | 97.08 | 37 | 6 |
| Shape | 1.18 | 44.30 | 564 | 85 |
| Realized design | 3.56 | 1.46 | 80424865 | 7050142 |

affect the quality of printed parts is the distance between the nozzle and the build platform [73]. This distance can impact the adhesion between the layers and can cause defects similar to those shown in Fig 19(C). It is essential to consider these factors during the design and optimization of 3D-printed parts to achieve the desired mechanical properties and avoid potential defects.

Based on the microscopic images, different kinds of voids can significantly impact the quality and properties of 3D-printed parts, and the infill patterns examined in this study is one such example [58]. The infill patterns, often referred to as meta-structures, play a crucial role in influencing the properties of the components. Additionally, there are other forms of anisotropy that can arise from changes in printing paths or inadequate print parameters. While these factors were not specifically investigated in this study, it was reported from the porosity analysis that the adhesion between the laminated layers has no significant role to play in flexural properties [6].

As illustrated in Figs 18 and 20, as well as Table 9, the length of the vertices and width of lines were measured in the microscopic images to visualize dimensional accuracy. Specifically, Fig 18 shows a schematic of the measured line width and line length.

The measured line width in this picture does not refer to the actual line width produced by the nozzle. Rather, it is the width of the walls in the structure, which may consist of multiple print paths.

Table 10 shows the measured values in an irregular octagon as a sample. The ImageJ software was used to measure the length and width of the lines, as well as the angle between them. The process repeated three times to ensure accuracy. The results were within a reasonable tolerance based on the difference between the demanded and measured values as well as the standard deviations according to Table 10. The comparison between the average values measured from the microscopic images and corresponding CAD models is done in this table. The purpose of this analysis was to assess how accurately the 3D printer was able to reproduce the intended geometry of the parts, and to identify any potential sources of error or variability in the printing process. By comparing the actual dimensions of the printed parts with the intended dimensions, it is possible to gain insight into the overall quality and reliability of the printing process, and to identify opportunities for improvement. A scatterplot was used to investigate the relationship between demanded and measured data (Fig 21). The rectangular sample with 40% of infill and the regular octagon with 60% of infill exhibited the highest deviations from the desired shapes.

Fig 21 illustrates the relationship between demanded and measured line widths and line lengths, with some alignment suggesting precision while others display noticeable deviations. Fig 21(A) compares the demanded and measured line widths, while Fig 21(B) does the same for lengths. In both of the presented plots, the data points fall within a range of 90% of the expected values. This issue indicates a high degree of consistency between the intended specifications and the actual measurements, reinforcing the reliability of the printing process in maintaining dimensional accuracy. The positive and linear relationship indicates that as the demanded line width or line length increases, there is a corresponding increase in the measured values, highlighting the consistent and proportional behavior between the two variables in the printing process.

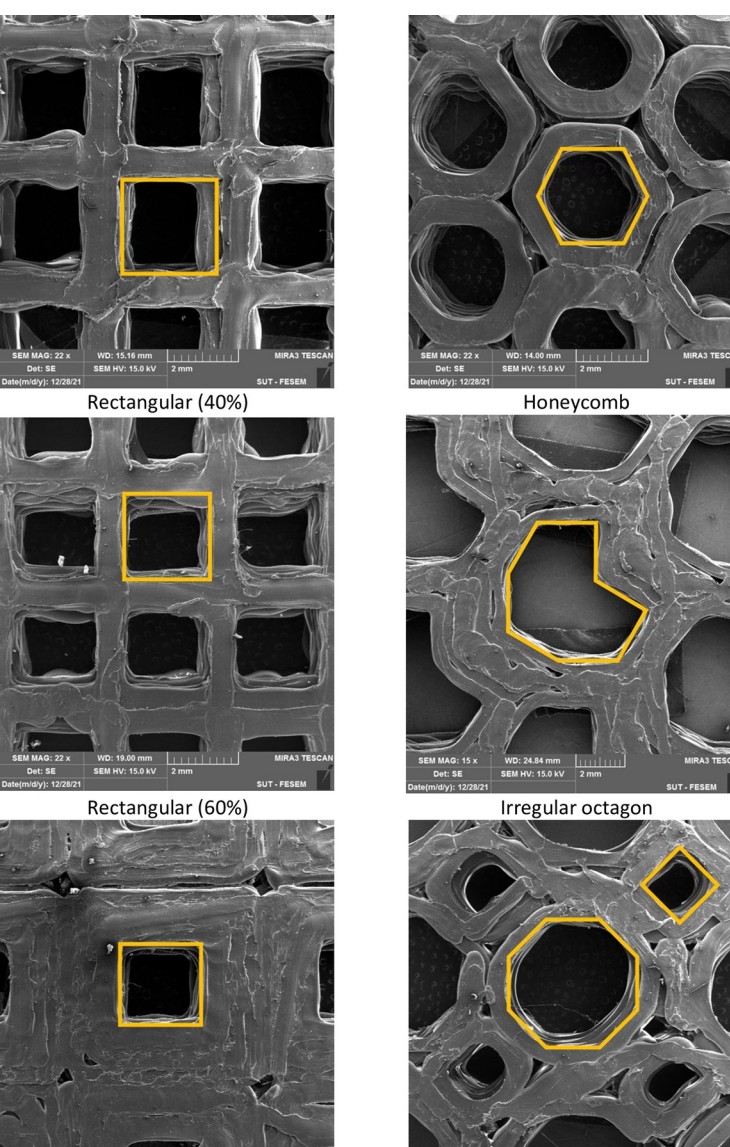

Rectangular (40%)

Honeycomb

Rectangular (60%)

Irregular octagon

Rectangular (80%)

Regular octagon

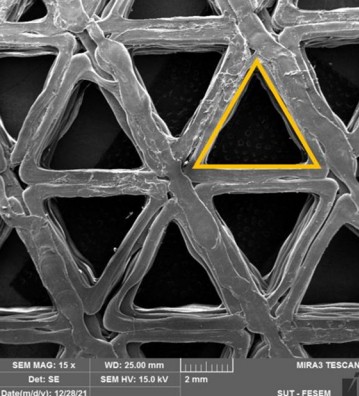

Triangular

**Fig 17. The low-magnification FESEM images of the printed samples.**

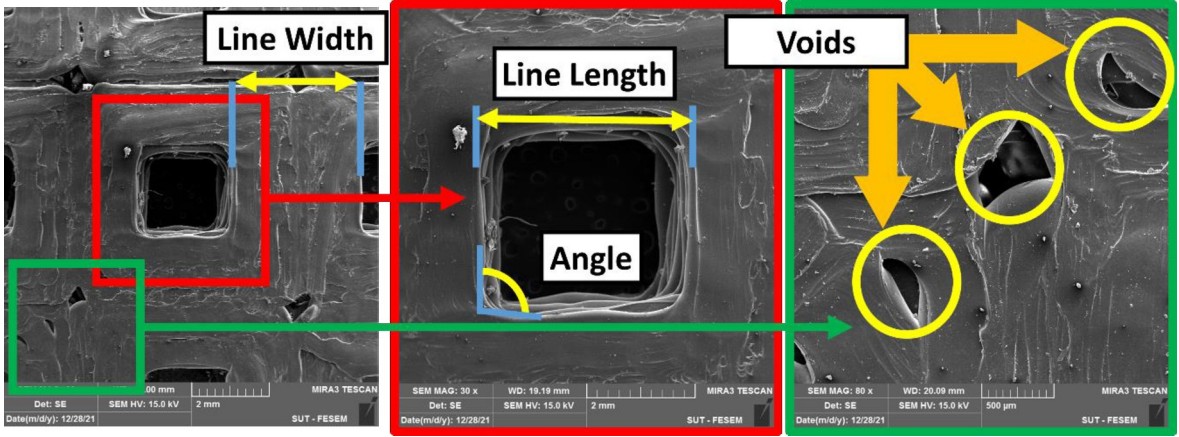

**Fig 18. The FESEM micrographs of the 3D-printed PLA with rectangular pattern and 80% of infill at different resolutions.**

Dimension reduction compared to the designed value occurs as a result of excessive deposition of the material during the extrusion process (due to not considering the dwell time at the endpoints of raster paths) and transverse flow (due to the relationship between temperature and viscosity). This is while the increase in dimensions can be due to the shrinkage of the printed polymer [74]. When there is an inaccuracy in the nozzle position, an imperfect configuration will occur. However, this issue could be solved by the calibration of the 3D printing machine [70]. It seems that such inaccuracy increases with the decrease in filling percentage and therefore the thickness of the walls. The sum of these items results in differences in filling densities and should be considered in the design phase.

Designing for high-performance in additive manufacturing, despite its limitations, requires exploring various unit cells and meta-structures to identify patterns that interact effectively with material behavior. In this research, the emphasis was primarily on numerical simulation and optimization of 3D-printed materials. The results were utilized to determine the parameters for experimental tests, the data of which has already been published [61, 62]. For further investigations, the following subjects are suggested,

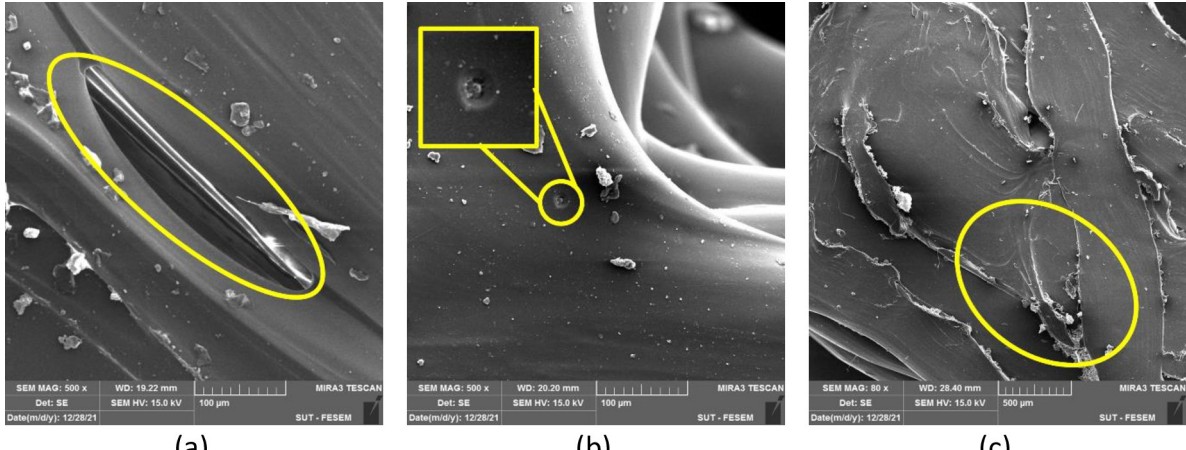

**Fig 19.** The defects observed in the samples with (a) 60% infill regular octagon, (b) 80% infill rectangular, and (c) 60% infill irregular octagon.

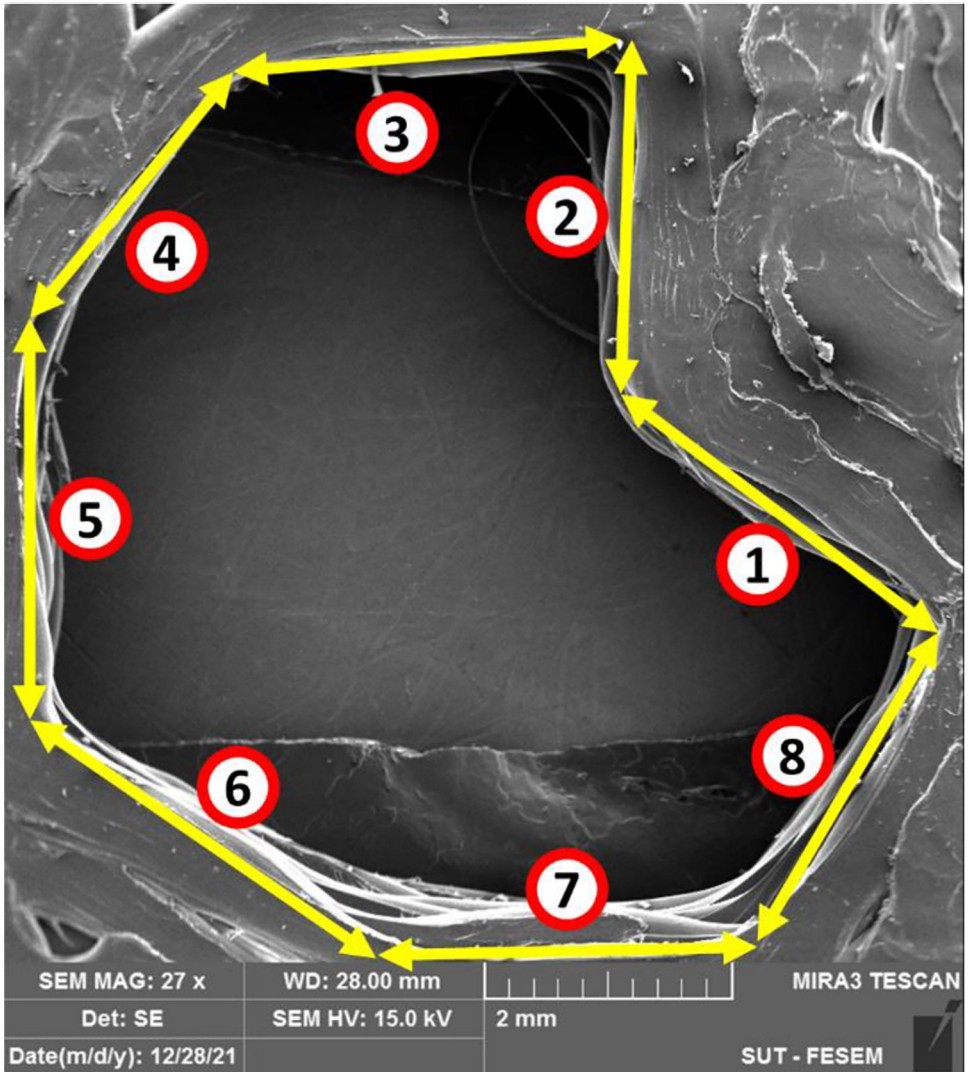

**Fig 20. The dimensional accuracy of the irregular octagon, measured from FESEM micrographs of the 3D-printed PLA with 60% of infill.**

**Table 9. The measured dimensions of the irregular octagon with 60% of infill.**

| ID | Length (mm) | Angle (degree) |
|----|-------------|----------------|
| 1  | 3.329       | 146.4          |
| 2  | 3.087       | 85.7           |
| 3  | 3.526       | 3.4            |
| 4  | 3.067       | 38.8           |
| 5  | 3.463       | 92.6           |
| 6  | 3.556       | 145.9          |
| 7  | 3.300       | 0.2            |
| 8  | 3.402       | 60.0           |

**Table 10. The dimensional variations of samples printed as per the experimental design.**

| Parameter / Fill pattern | Rectangular | | | Triangular | Honeycomb | Regular octagon | Irregular octagon |
|---|---|---|---|---|---|---|---|
| Demanded Infill (%) | 40 | 60 | 80 | 60 | 60 | 60 | 60 |
| Measured Infill (%) | 74.61 | 77.44 | 90.27 | 69.7 | 75.55 | 70.24 | 71.66 |
| Standard Deviation | 86.53 | 29.07 | 12.84 | 16.17 | 25.92 | 17.07 | 19.43 |
| Demanded Line Width (%) | 2.75 | 2.75 | 2.75 | 4.50 | 1.25 | 1.50 | 2.50 |
| Measured Line Width (%) | 2.78 | 2.57 | 2.41 | 4.36 | 1.49 | 1.54 | 2.39 |
| Standard Deviation | 0.12 | 0.13 | 0.09 | 0.17 | 0.10 | 0.10 | 0.15 |
| Demanded Line Length (mm) | 0.50 | 1.25 | 3.00 | 1.00 | 1.60 | 1.25 | 3.3 |
| Measured Line Length (mm) | 0.83 | 1.18 | 3.41 | 1.02 | 1.57 | 1.46 | 3.39 |
| Standard Deviation | 0.10 | 0.14 | 0.13 | 0.05 | 0.11 | 0.22 | 0.09 |

- Path planning and defining the route that the print head takes

- Providing topology optimization algorithms to achieve desired properties for specific applications

- Using exactly the material properties of the meta-structure and anisotropy (considering the 3D printing factors) for the lifetime prediction

- Examine the fatigue lifetime experimentally for the topology-optimized samples.

## 4) Conclusions

This research used multi-objective optimization to determine optimal parameters for PLA cylinders, considering fatigue lifetime, mass, and infill patterns. Structural topology optimization was conducted using a two-objective algorithm. Additionally, FESEM analysis ensured dimensional accuracy, and the influence of inherent defects in the FDM process was considered. The results are listed below:

- The stress analysis results showed that although the triangular pattern had the lowest mass in all cases, irregular octagon, rectangular and regular octagon had the lowest stresses in 40%, 60%, and 80% of the infill, respectively.

- The study introduced three sets of objective functions for design purposes, considering mass changes, fatigue lifetime changes, and the distance from the optimal point. The optimal solution was found to be the horizontally printed rectangular filling pattern with a 60% infill density. This pattern achieved a 39% reduction in weight while experiencing only a 22% decrease in fatigue lifetime. The mass reduction was determined through software analysis of designed CAD models, while the infill density was assessed using the ImageJ software.

- Topology optimization in Dassault Systems software proposed a metamaterial structure to minimize mass and strain energy under stress constraints. Achieved results: mass of 1.1767g, strain energy of 0.056 J (after 50 cycles), maximum von-Mises stress of 97.08 MPa. Microscopic inspections studied manufacturing defects and their impact on topology optimization.

- Shape optimization was performed to reduce stress concentration by displacing boundary nodes. Maximum von-Mises stress decreased to 44.30 MPa after 15 design cycles, leading to increased fatigue lifetime by 527 and 79 cycles compared to topology optimization in vertical and horizontal samples, respectively.

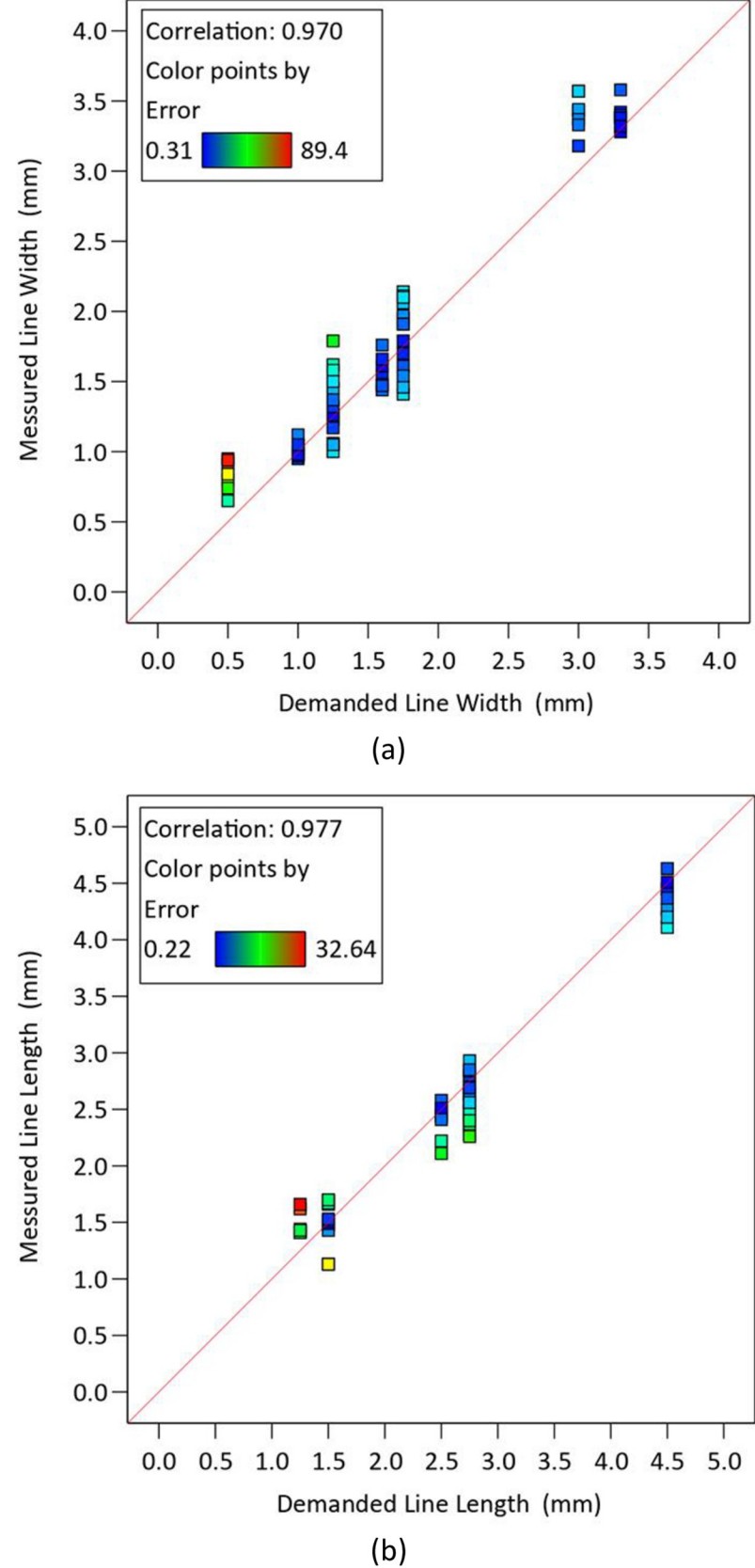

**Fig 21.** The scatter plot of demanded values versus the measured data for (a) the line length and (b) the line width.

- Finally, a design realization step was performed, and a new geometry was introduced that has 96.70% lower stress and 201.69% higher mass compared with topology-optimized geometry. This is while this geometry is lighter than 60% infill samples with different patterns.

- Based on FESEM images, the rectangular pattern has the best regularity, while the regular octagon pattern was the most irregular.

- The line spacing is significant, especially in the honeycomb pattern, and should be controlled by printing temperature.

- The scatterplot shows a strong positive and linear relationship between the demanded and measured line length and line width.

- Inaccuracy in the position of the nozzle increases with the decrease in filling percentage and, therefore, the thickness of the walls. However, the rectangular sample with 40% infill and regular octagon with 60% infill was the worst parameters.

## Supporting information

**S1 Fig. The GIF file for the topology optimization during the process.**
(GIF)

**S1 Graphical abstract.**
(TIF)

## Author Contributions

**Conceptualization:** Mohammad Azadi.

**Data curation:** Ali Dadashi.

**Formal analysis:** Ali Dadashi.

**Funding acquisition:** Mohammad Azadi.

**Investigation:** Ali Dadashi, Mohammad Azadi.

**Methodology:** Ali Dadashi, Mohammad Azadi.

**Project administration:** Mohammad Azadi.

**Resources:** Ali Dadashi, Mohammad Azadi.

**Software:** Ali Dadashi.

**Supervision:** Mohammad Azadi.

**Validation:** Ali Dadashi.

**Visualization:** Ali Dadashi.

**Writing – original draft:** Ali Dadashi.

**Writing – review & editing:** Mohammad Azadi.

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
