## [Decision Letter · Decision Letter 0]

3 Apr 2023

PONE-D-23-02761Multi-objective numerical optimization of 3D-printed polylactic acid bio-metamaterial based on topology, filling pattern, and infill density via fatigue lifetime and weightPLOS ONE

Dear Dr. Azadi,

Thank you for submitting your manuscript to PLOS ONE. After careful consideration, we feel that it has merit but does not fully meet PLOS ONE’s publication criteria as it currently stands. Therefore, we invite you to submit a revised version of the manuscript that addresses the points raised during the review process.

Please, address all the comments made by the reviewers, specially those related to the design of experiments and the mechanical modeling. 

We look forward to receiving your revised manuscript.

Kind regards,

Antonio Riveiro Rodríguez, PhD

Academic Editor

PLOS ONE

“The authors would like to acknowledge the financial support of the Iran Small Industries and Industrial Parks Organization (ISIPO) for this project under grant number of 23607.”

“The authors would like to acknowledge the financial support of the Iran Small Industries and Industrial Parks Organization (ISIPO) for this project under grant number of 23607.”

6. We note that you have indicated that data from this study are available upon request. PLOS only allows data to be available upon request if there are legal or ethical restrictions on sharing data publicly. For more information on unacceptable data access restrictions, please see http://journals.plos.org/plosone/s/data-availability#loc-unacceptable-data-access-restrictions.

Reviewers' comments:

Reviewer's Responses to Questions

**Comments to the Author**

1. Is the manuscript technically sound, and do the data support the conclusions?

Reviewer #1: Partly

Reviewer #2: Partly

Reviewer #3: Yes

Reviewer #4: Yes

Reviewer #5: Yes

Reviewer #6: No

2. Has the statistical analysis been performed appropriately and rigorously? 

Reviewer #1: Yes

Reviewer #2: Yes

Reviewer #3: Yes

Reviewer #4: Yes

Reviewer #5: Yes

Reviewer #6: No

3. Have the authors made all data underlying the findings in their manuscript fully available?

Reviewer #1: No

Reviewer #2: No

Reviewer #3: Yes

Reviewer #4: Yes

Reviewer #5: Yes

Reviewer #6: No

4. Is the manuscript presented in an intelligible fashion and written in standard English?

Reviewer #1: Yes

Reviewer #2: Yes

Reviewer #3: Yes

Reviewer #4: Yes

Reviewer #5: Yes

Reviewer #6: No

5. Review Comments to the Author

Reviewer #1: Dear Authors,

Your paper is interested and well structured, however some questions arose during the review process:

1. Introduction: More extensive presentation of Pareto optimisation needs to be done with emphasis on the research in the presented research area.

2. Page 3 (P3): works are also done presenting the compressive loading of the PLA printed material - consider DOI: 10.3390/ma13194456

3. Recently a review of the influence of printing parameters on mechanical properties was published: consider their findings: https://doi.org/10.3390/polym15030716

4, Material and material properties - you mention two references. However, are all those material properties from your table 1 experimentally determined?

5. In chapter 2-2 the test specimen also numerically verified is also dimensional presented. It has diameter of 9 mm with a shell of 0,4 mm on its surface. How you can then claim that the element size of 0.7 mm is optimal for your evaluations. This is highly doubtful and needs further explanation. Additionally, with this ratio of specimen diameter versus element size you have only 13 elements on the cross-section. Finally, the selection of the element size needs to be discussed more into the detail.

6. Figure 2 - font in the upper row is too large.

7. Figure 5 presents the used machine for specimen printing. The scale is missing on the figure. The data about the printer's accuracy is missing.

8. It is noted that the samples have been printed 5x larger as the ones used in the CAD model.Why. This needs to be explained into the detail. Furthermore, the comment about the influence of this scaling is missing.

9. Table 7: first column is too narrow and cuts the text - correct this. Also in this table, you presents the Von Mises stress distributions. Can you compare them also with the experimental data? From this comparison it is also not evident what is compressive and what tensile loaded. This need to be corrected and need also further explanation.

10. Numerical simulation: it is obligatory to present the used FE mesh for all shapes used in the comparative evaluations. Furthermore, you claims in the text that some parts are compressive and some tensile loaded. Are the tensile and compressive material properties the same. The simulations of the compressive printed parts were done by Karimi et al in https://doi.org/10.3390/app12168018 where also influence of the printed shape of the particular layer was analysed. What shape of the particular layer have you selected? Perpendicular of the real one?

The Pareto optimisation is done correct and there I have no comments.

11. Figure 11 - again only the Von Mises stress is presented. Why?

12. Figure 12 - the scales needs to be larger and more readable. From this figure it is also evident that the printing quality is not good. How this quality influence the properties of the real part and even more important - how you can implement those errors in the numerical simulations.

13. Fugure 13 - the measured distances and angles are vague and not accurate. Comment such results.

14. Conclusions: you mention the weight after 50 cycles. Does this changes through the cyclic loading? Furthermore - how is this weight comparable to the ones calculated from the geometry of the CAD model with corresponding material density. What is the real material density of your used material?

Reviewer #2: Dear Authors,

in your interesting manuscript, the following points should be added/changed to further improve it:

- Introduction: "The robust objective function was weighted mean and variance of the deterministic objective" - what is meant here?

- Section 2.2: Why were these infill patterns chosen and not, e.g., gyroid, which was found most suitable in experiments dealing with PLA as shape memory polymer?

- Section 2.2: I assume the "thickness of 0.4 mm for each layer" is actually a width of 0.4 mm; else the layer thickness would be identical to the typical nozzle width.

- Table 4: What is the difference between "all data" and "mean value"?

- Table 5: All "100" must be "100%". Please think about renaming "Delta W" to "Delta W/W_solid" (ditto for Delta N_f) as your Deltas are relative deviations given in %, while a Delta value is normally assumed to be an absolute difference.

- Eqs. 8-11 do not look logically correlated; please check again where you have 1, 2, M, M-1 (or M+1).

- Fig. 4: What do the green lines mean?

- "It should be noted that due to the complex shapes, samples were printed 5 times larger than CAD models based on the

capability of machine." - why? Which dimensions do the samples have and how many filled top/bottom layers and perimeters? And why was no commercially available printer used?

- Fig. 6b: The y-axis shows the mass, not the weight of the samples. What do you mean in the corresponding text with "unpredictable relationship" - actually this is quite clear and logical.

- Fig. 7: Where is the difference between blue and green points?

- Fig. 9: Does this optimized structure take the strong anisotropies in FDM printing into account?

- Fig. 11: How can the optimized shape be FDM printed?

- Table 10: The second column is "Mass (g)".

- Fig. 12: How are the air voids visible here taken into account in the simulation?

- Fig. 14: Which sorts of defects do you see in (b) and (c)? At least there are no air voids.

- Fig. 15: The length unit is wrong. Length and angle values have much too high accuracy; you can definitely not measure with 6 significant digits accuracy on such an image.

- Table 11: Where do these values stem from? Where are the units? Where do the standard deviations belong to? How should a printed line, typically of 0.4 mm width, have a demanded width of 2.75 mm etc., and what is meant with the line length? Besides, it would be good to correlate the measured infills (surely not measured with this accuracy) with the measured sample masses.

- "Based on the scatterplot in Figure 16, a strong positive and linear relationship between the two variables" - which ones?

- Fig. 16: all texts are too small to be readable.

- Conclusion: The information that a sample with infill density 60% is 39% lighter than the 100% sample sounds logical, but does not fit to the measured infill percentages according to Table 11. It is thus really necessary to defined how you measured this value.

- The conclusion is too long.

Reviewer #3: The author has done good work. The work is related to “multi-objective numerical optimization of 3D-printed polylactic acid bio-metamaterial based on topology, filling pattern, and infill density via fatigue lifetime and weight” and has provided good insights on fatigue life time for the Additively manufactured printed prototypes. Here are some suggestions which the author can incorporate in the manuscript for improvement in the output of this work.

1. Author should provide scope of the work and literature gap as separate heading for better understanding

2. What is bio-metamaterial? Bio material is separate term whereas Meta material is something related to design-based objective of materials.

3. Figure 6a shows the 100% infill density part has shown stress capacity <10MPa. But the original value for PLA material lies somewhere between 30-70MPa. Why the predicted value is many times lower than the actual value?

4. Whereas Figure 6b is not important as it is well established fact that when density increases the weight will increase.

5. No practical work has been performed other than mathematical relations and numerical simulation. You should 3D print any material and test it with your optimized condition and verify the predicted model using real environment condition?

6. One real 3D printed object based on optimized or suggested model should be printed for PLA material and the model should be verified to find the accuracy of the model.

7. Previous literature work or result should be compared with the proposed model result for validation purpose along with suitable graph of validation.

8. Following literature may be added to enhance the literature review

1. Kumar S, Singh I, Kumar D, Yahya MY, Rahimian Koloor SS. Mechanical and Morphological Characterizations of Laminated Object Manufactured 3D Printed Biodegradable Poly (lactic) acid with Various Physical Configurations. Journal of Marine Science and Engineering. 2022 Dec;10(12):1954.

2. Kumar S, Singh I, R. Koloor SS, Kumar D, Yahya MY. On Laminated Object Manufactured FDM-Printed ABS/TPU Multimaterial Specimens: An Insight into Mechanical and Morphological Characteristics. Polymers. 2022 Sep 28;14(19):4066.

3. Kumar S, Singh R, Singh TP, Batish A. Comparison of mechanical and morphological properties of 3-D printed functional prototypes: Multi and hybrid blended thermoplastic matrix. Journal of Thermoplastic Composite Materials. 2022 May;35(5):692-707.

4. Kumar S, Singh R, Batish A, Singh TP. Additive manufacturing of smart materials exhibiting 4-D properties: a state of art review. Journal of Thermoplastic Composite Materials. 2022 Sep;35(9):1358-81.

Reviewer #4: The manuscript represents good potentials for readers .

Al section represents technically correct data.

Analysis is technically sound.

Data is supported by relevant literature.

Authors have taken nice efforts to fulfil requirements of Journal.

Reviewer #5: The study is well conducted and results have been theoretically discussed. This manuscript could be accepted in its current form. The execution of the experimentation and testing is at par with the current research requisites.

Reviewer #6: The focus of this work is studying optimization of 3D-printed polylactic. The present work does not include any novel materials or methods. This manuscript is insufficient of novelty or significant contribution for publication as all materials and methods have been published in the last 3 years, see the references. In conclusion, the innovation of the manuscript is very scarce.

Apart from the lack of contribution, it has two major issues that are lack of any design of experiments for 3D printing process and loading in a cyclic manner for this particular application. That is why we see quick failure under loading. I would strongly suggest the authors to carry out a proper design of experiments to see the effects of printing parameters like temperature of nozzle, speed of print head, fibre orientation, layers arrangement, infill pattern, etc. on the performance of the structure under mechanical loading. As it is under cyclic loading in the real application, the mechanical loading modelling must be cyclic as well.

A new version of the manuscript with all major fundamental changes could be re-submitted for review.

6. PLOS authors have the option to publish the peer review history of their article (what does this mean?). If published, this will include your full peer review and any attached files.

Reviewer #1: No

Reviewer #2: No

Reviewer #3: No

Reviewer #4: **Yes: **Sunil Jaysing Raykar

Reviewer #5: No

Reviewer #6: No

---

## [Author Response · Author response to Decision Letter 0]

10 Jun 2023

Please check the file uploaded for answers to the reviewers' comments.

---

## [Decision Letter · Decision Letter 1]

12 Jul 2023

PONE-D-23-02761R1Multi-objective numerical optimization of 3D-printed polylactic acid bio-metamaterial based on topology, filling pattern, and infill density via fatigue lifetime and massPLOS ONE

Dear Dr. Azadi,

Thank you for submitting your manuscript to PLOS ONE. After careful consideration, we feel that it has merit but does not fully meet PLOS ONE’s publication criteria as it currently stands. Therefore, we invite you to submit a revised version of the manuscript that addresses the points raised during the review process.

Please, address all the comments provided in the manuscript by reviewer 1. 

We look forward to receiving your revised manuscript.

Kind regards,

Antonio Riveiro Rodríguez, PhD

Academic Editor

PLOS ONE

Journal Requirements:

Reviewers' comments:

Reviewer's Responses to Questions

**Comments to the Author**

1. If the authors have adequately addressed your comments raised in a previous round of review and you feel that this manuscript is now acceptable for publication, you may indicate that here to bypass the “Comments to the Author” section, enter your conflict of interest statement in the “Confidential to Editor” section, and submit your "Accept" recommendation.

Reviewer #1: (No Response)

Reviewer #2: (No Response)

2. Is the manuscript technically sound, and do the data support the conclusions?

Reviewer #1: Yes

Reviewer #2: (No Response)

3. Has the statistical analysis been performed appropriately and rigorously? 

Reviewer #1: Yes

Reviewer #2: (No Response)

4. Have the authors made all data underlying the findings in their manuscript fully available?

Reviewer #1: Yes

Reviewer #2: (No Response)

5. Is the manuscript presented in an intelligible fashion and written in standard English?

Reviewer #1: No

Reviewer #2: (No Response)

6. Review Comments to the Author

Reviewer #1: Dear authors,

you have improved the manuscript. However, there are still some opened questions which need to be resolved prior to the publication. Also the English language needs to be improved.

Reviewer #2: (No Response)

7. PLOS authors have the option to publish the peer review history of their article (what does this mean?). If published, this will include your full peer review and any attached files.

Reviewer #1: No

Reviewer #2: No

---

## [Author Response · Author response to Decision Letter 1]

2 Aug 2023

Please check the file for answers to comments.

---

## [Decision Letter · Decision Letter 2]

21 Aug 2023

Multi-objective numerical optimization of 3D-printed polylactic acid bio-metamaterial based on topology, filling pattern, and infill density via fatigue lifetime and mass

PONE-D-23-02761R2

Dear Dr. Azadi,

We’re pleased to inform you that your manuscript has been judged scientifically suitable for publication and will be formally accepted for publication once it meets all outstanding technical requirements.

Kind regards,

Antonio Riveiro Rodríguez, PhD

Academic Editor

PLOS ONE

Reviewers' comments:

Reviewer's Responses to Questions

**Comments to the Author**

1. If the authors have adequately addressed your comments raised in a previous round of review and you feel that this manuscript is now acceptable for publication, you may indicate that here to bypass the “Comments to the Author” section, enter your conflict of interest statement in the “Confidential to Editor” section, and submit your "Accept" recommendation.

Reviewer #1: All comments have been addressed

Reviewer #2: (No Response)

2. Is the manuscript technically sound, and do the data support the conclusions?

Reviewer #1: Yes

Reviewer #2: (No Response)

3. Has the statistical analysis been performed appropriately and rigorously? 

Reviewer #1: Yes

Reviewer #2: (No Response)

4. Have the authors made all data underlying the findings in their manuscript fully available?

Reviewer #1: Yes

Reviewer #2: (No Response)

5. Is the manuscript presented in an intelligible fashion and written in standard English?

Reviewer #1: Yes

Reviewer #2: (No Response)

6. Review Comments to the Author

Reviewer #1: Dear Authors,

your manuscript was improved to the level where I can suggest it for publication. However, prior to that I need to address two typo errors regarding the Figures 6 and 18. There the diagrams (Fig6) and figure title (Fig 18) are not in accordance with corresponding text. Take a carefully look of the attached file. This is obligatory to improve for quality manuscript.

Reviewer #2: (No Response)

7. PLOS authors have the option to publish the peer review history of their article (what does this mean?). If published, this will include your full peer review and any attached files.

Reviewer #1: No

Reviewer #2: No

---

## [Editor Report · Acceptance letter]

18 Sep 2023

PONE-D-23-02761R2 

Multi-objective numerical optimization of 3D-printed polylactic acid bio-metamaterial based on topology, filling pattern, and infill density via fatigue lifetime and mass 

Dear Dr. Azadi:

I'm pleased to inform you that your manuscript has been deemed suitable for publication in PLOS ONE. Congratulations! Your manuscript is now with our production department. 

Kind regards, 

on behalf of

Dr. Antonio Riveiro Rodríguez 

Academic Editor

PLOS ONE